# Simplified Graph Convolution with Heterophily

**Sudhanshu Chanpuriya**
University of Massachusetts Amherst
schanpuriya@cs.umass.edu

**Cameron Musco**
University of Massachusetts Amherst
cmusco@cs.umass.edu

## Abstract

Recent work has shown that a simple, fast method called Simple Graph Convolution (SGC) (Wu et al., 2019), which eschews deep learning, is competitive with deep methods like graph convolutional networks (GCNs) (Kipf & Welling, 2017) in common graph machine learning benchmarks. The use of graph data in SGC implicitly assumes the common but not universal graph characteristic of homophily, wherein nodes link to nodes which are similar. Here we confirm that SGC is indeed ineffective for heterophilous (i.e., non-homophilous) graphs via experiments on synthetic and real-world datasets. We propose Adaptive Simple Graph Convolution (ASGC), which we show can adapt to both homophilous and heterophilous graph structure. Like SGC, ASGC is not a deep model, and hence is fast, scalable, and interpretable; further, we can prove performance guarantees on natural synthetic data models. Empirically, ASGC is often competitive with recent deep models at node classification on a benchmark of real-world datasets. The SGC paper questioned whether the complexity of graph neural networks is warranted for common graph problems involving homophilous networks; our results similarly suggest that, while deep learning often achieves the highest performance, heterophilous structure alone does not necessitate these more involved methods.

## 1 Introduction

Data involving relationships between entities arises in biology, sociology, and many other fields, and such data is often best expressed as a graph. Therefore, models of graph data that yield algorithms for common graph machine learning tasks, like node classification, have wide-reaching impact. Deep learning (LeCun et al., 2015) has enjoyed great success in modeling image and text data, and graph convolutional networks (GCNs) (Kipf & Welling, 2017) attempt to extend this success to graph data. Various deep models branch off from GCNs, offering additional speed, accuracy, or other features. However, like other deep models, these algorithms involve repeated nonlinear transformations of inputs and are therefore time and memory intensive to train.

Recent work has shown that a much simpler model, simple graph convolution (SGC) (Wu et al., 2019), is competitive with GCNs in common graph machine learning benchmarks. SGC is much faster than GCNs, partly because the role of the graph in SGC is restricted to a feature extraction step in which node features are smoothed across the graph; by contrast, GCNs include graph convolution steps as part of an end-to-end model, resulting in expensive backpropagation calculations. However, the feature smoothing operation in SGC implicity assumes the common but not universal graph characteristic of homophily, wherein nodes mostly link to similar nodes; indeed, recent work (NT & Maehara, 2019) suggests that many GCNs may assume such structure. We ask whether a feature extraction approach, that, like SGC, is free of deep learning, can tackle heterophilous (i.e., non-homophilous) graph structure. We see this work as extending the following broader research question introduced by the SGC paper to a wider range of graphs:

> *Are nonlinearities, end-to-end backpropagation, and other characteristics of deep learning essential to effective learning on graphs?*

36th Conference on Neural Information Processing Systems (NeurIPS 2022).

**Contributions**  We confirm that SGC, which uses a fixed smoothing filter, can indeed be ineffective for heterophilous features via experiments on synthetic and real-world datasets. We propose adaptive simple graph convolution (ASGC), which fits a different filter for each feature. These filters can be smoothing or non-smoothing, and thus can adapt to both homophilous and heterophilous structures. Like SGC, ASGC is not a deep model, instead being based on linear least squares, and hence is fast, scalable, and interpretable. We propose a natural synthetic model for networks with a node feature, Featured Stochastic Block Models (FSBMs), and prove that ASGC denoises the network feature regardless of whether the model is set to produce homophilous or heterophilous networks, in contrast to SGC, which we show is inappropriate for the heterophilous setting. Finally, we show that the performance of ASGC is superior to that of SGC at node classification on real-world heterophilous networks, and generally competitive with recent deep methods on a benchmark of both heterophilous and homophilous networks. The SGC paper suggested that deep learning is not necessarily required for good performance on common graph learning tasks and benchmarks involving homophilous networks; our results suggest that simple methods can also be competitive for heterophilous networks.

## 2   Background

**Preliminaries**  We first establish common notation for working with graphs. An undirected, possibly weighted graph on $n$ nodes can be represented by its symmetric adjacency matrix $\boldsymbol{A} \in \mathbb{R}_{\geq 0}^{n \times n}$. Letting $\mathbf{1}$ denote an all-ones column vector, $\boldsymbol{d} = \boldsymbol{A}\mathbf{1}$ is the vector of degrees of the nodes; the degree matrix $\boldsymbol{D}$ is the diagonal matrix with $\boldsymbol{d}$ along the diagonal. The normalized adjacency matrix is given by $\boldsymbol{S} = \boldsymbol{D}^{-1/2}\boldsymbol{A}\boldsymbol{D}^{-1/2}$, while the symmetric normalized graph Laplacian is $\boldsymbol{L} = \boldsymbol{I} - \boldsymbol{S}$, where $\boldsymbol{I}$ is an identity matrix. Note that the eigenvectors of $\boldsymbol{L}$ are exactly the eigenvectors of $\boldsymbol{S}$. It is a well known fact that the eigenvalues of $\boldsymbol{S}$ are contained within $[-1, +1]$. It follows that the eigenvalues of $\boldsymbol{L}$ are within $[0, 2]$, so $\boldsymbol{L}$ is positive semidefinite.

Consider a feature $\boldsymbol{x} \in \mathbb{R}^n$ on the graph, that is, a real-valued vector where each entry is associated with a node. The quadratic form over $\boldsymbol{L}$ is known to have the following equivalence:

$$\boldsymbol{x}^\top \boldsymbol{L} \boldsymbol{x} = \tfrac{1}{2} \sum\nolimits_{(i,j) \in [n]^2} A_{ij} \left( \tfrac{1}{\sqrt{d_i}} x_i - \tfrac{1}{\sqrt{d_j}} x_j \right)^2 .$$

Up to a reweighting based on the nodes' degrees, this expression is the sum of squared differences of the feature's values between adjacent nodes. Hence, the quadratic form has a low value, near $0$, if the feature $\boldsymbol{x}$ is 'smooth' across the graph, that is, if adjacent nodes have generally similar values of the feature. Similarly, the quadratic has a high value if the feature is 'rough' across the graph, if adjacent nodes have generally differing values of the feature. When $\boldsymbol{x}$ is an eigenvector of $\boldsymbol{L}$, these 'smooth' and 'rough' cases correspond to the eigenvalue being low or high, respectively. (In terms of eigenvalues of $\boldsymbol{S}$, the opposite is true, with positive eigenvalues being smooth and negative eigenvalues being rough.) More generally, decomposition of an arbitrary feature vector $\boldsymbol{x}$ as a linear combination of the eigenvectors of $\boldsymbol{L}$ separates $\boldsymbol{x}$ into components ranging from 'smooth' to 'rough' across the graph. In the graph signal processing literature (Ortega et al., 2018; Huang et al., 2016; Stanković et al., 2019), these smooth and rough components are also called low and high frequency 'modes,' respectively, based on the eigenvalues of $\boldsymbol{L}$.

**Simple graph convolution**  Our work is primarily inspired by the simple graph convolution (SGC) algorithm (Wu et al., 2019). SGC comprises logistic regression after the following feature extraction step, which can be interpreted as smoothing the nodes' features across the graph's edges:

$$\boldsymbol{x}_{\mathrm{SGC}} = \tilde{\boldsymbol{S}}^K \boldsymbol{x}. \tag{1}$$

This equation shows how a single raw feature $\mathbf{x}$ is filtered to produce the smoothed feature $\boldsymbol{x}_{\mathrm{SGC}}$. Here $\tilde{\boldsymbol{S}} \in \mathbb{R}^{n \times n}$ is the normalized adjacency matrix after addition of a self-loop to each node (that is, addition of the identity matrix to the adjacency matrix), and $K \in \mathbb{Z}_+$ is a hyperparameter; $K$ determines the radius of the filter, that is, the maximum number of hops between two nodes whose features can directly influence others' features in the filtering process. Wu et al. (2019) show that the addition of the self-loop increases the minimum eigenvalue of $\boldsymbol{S}$; intuitively, the self-loop limits the extent to which a feature can be 'rough' across the graph. This results in the highest magnitude eigenvalues of the normalized adjacency $\tilde{\boldsymbol{S}}$ tending to be positive (smooth). Because the eigenvalues of $\tilde{\boldsymbol{S}}$ all have magnitude at most $1$, powering up $\tilde{\boldsymbol{S}}$ results in a filter which generally attenuates the

feature $x$, but does so least along these high magnitude, smooth eigenvectors. Hence the SGC filter smooths out the feature locally along the edges. Since it attenuates the high-frequency modes more than the low-frequency ones, SGC is described as a 'low-pass' filter.

**Heterophily**    If node features are used for node classification or regression, smoothing the features of nodes along edges encourages similar predictions along locally connected nodes. This seems sensible when locally connected nodes should be generally similar in terms of features and labels; if the variance in features and labels between connected nodes is generally attributable to noise, then this smoothing procedure acts as a useful denoising step. Graphs in which connected nodes tend to be similar are called homophilous or assortative. An example would be a citation network of papers on various topics: papers concerning the same topic tend to cite each other. Much of the existing work on graph models has an underlying assumption of network homophily, and there has been significant recent interest (discussed further in Section 6) on the limitations of graph models at addressing network heterophily/disassortativity, wherein connections tend to be between dissimilar nodes. An example would be a network based on adjacencies of words in text, where the labels are based on the part of speech: adjacent words tend to be of different parts of speech. For disassortative networks, smoothing a feature across connections as in SGC may not be sensible, since encouraging predictions of connected nodes to be similar is contrary to disassortativity.

## 3    Methodology

**Adaptive SGC**    In our method, we replace the fixed feature propagation step of SGC (Equation 1) with an adaptive one, which may or may not be smoothing based on the feature and graph. We produce a filtered version $x_{\text{ASGC}}$ of the raw feature $x$ by multiplying $x$ with a learned polynomial of the normalized adjacency matrix $S$; this polynomial is set so that $x_{\text{ASGC}} \approx x$:

$$x_{\text{ASGC}} = \left( \sum\nolimits_{k=0}^{K} \beta_k S^k \right) x \approx x, \tag{2}$$

where $K$ is a hyperparameter which, as in SGC, controls the radius of feature propagation, and the coefficients $\beta \in \mathbb{R}^{K+1}$ are learned by minimizing the approximation error in a least squares sense. Note that, unlike SGC, we do not add self-loops to $S$. This approximation error would be trivially minimized with $\beta_0 = 1$ and all other coefficients set to zero, resulting in $x_{\text{ASGC}} = S^0 x = x$, so we regularize the magnitude of $\beta_0$.

More concretely, let $T \in \mathbb{R}^{n \times (K+1)}$ denote the Krylov matrix generated by multiplying a feature vector $x \in \mathbb{R}^n$ by the normalized adjacency matrix $S$ up to some $K \in \mathbb{Z}_{>0}$ times:

$$T = \left( S^0 x; S^1 x; S^2 x; \ldots; S^K x \right). \tag{3}$$

Here the leftmost column of $T$ is just the raw feature $x$, and each column represents the feature generated by propagating the feature vector to its left across the graph, i.e., by multiplying once by $S$. We produce a filtered version $x_{\text{ASGC}}$ of the raw feature $x$ by linear combination of the columns of $T$. That is, $x_{\text{ASGC}} = T\beta$, and we set the combination coefficients $\beta \in \mathbb{R}^{k+1}$ by minimizing a loss function as follows:

$$\min_{\beta} \left( \|T\beta - x\|_2^2 + (R\beta_0)^2 \right). \tag{4}$$

The term on the right is $L_2$ regularization applied to the zeroth combination coefficient $\beta_0$ (which multiplies the raw feature $S^0 x$), and $R \in \mathbb{R}_{\geq 0}$ if a hyperparameter which controls the strength of this regularization. Equation 4 is solved for the optimal coefficents $\beta$ using linear least squares.

**Intuition**    As noted above, if we set the regularization $R = 0$, or more generally in the limit as $R \to 0$, the approximation error in Equation 4 is trivially minimized and results in $x_{\text{ASGC}} = x$; that is, the learned filter just ignores the graph structure and maps the input feature through unchanged. Nonzero regularization forces the least squares reconstruction to use the graph structure as it approximates the raw, unpropagated feature. Ideally, this results in a denoising effect that is able to extend beyond SGC's fixed smoothing along edges. For example, when a graph $S$ is homophilous with

---
**Algorithm 1** Adaptive Simple Graph Convolution (ASGC) Filter
---
**Input:** undirected graph $\boldsymbol{A} \in \{0,1\}^{n \times n}$, node feature $\boldsymbol{x} \in \mathbb{R}^n$,
        number of hops $K \in \mathbb{Z}_{>0}$, regularization strength $R \in \mathbb{R}_{\geq 0}$
**Output:** ASGC-filtered feature $\boldsymbol{x}_{\text{ASGC}}$
$\boldsymbol{D} \leftarrow \text{diag}\left(\boldsymbol{A}\boldsymbol{1}\right)$ {degree matrix}
$\boldsymbol{S} \leftarrow \boldsymbol{D}^{-1/2}\boldsymbol{A}\boldsymbol{D}^{-1/2}$ {normalized adjacency matrix}
$\boldsymbol{T} \in \mathbb{R}^{n \times (K+1)} \leftarrow \boldsymbol{0}$ {$k$-step propagated features}
$\boldsymbol{T}_0 \leftarrow \boldsymbol{x}$
**for** $k = 1$ **to** $K$ **do**
     $\boldsymbol{T}_k \leftarrow \boldsymbol{S}\boldsymbol{T}_{k-1}$
**end for**
$\boldsymbol{R} \in \mathbb{R}^{K+1} \leftarrow (R, 0, 0, \ldots, 0)$
$\boldsymbol{\beta} \leftarrow$ least squares solution of $\binom{\boldsymbol{T}}{\boldsymbol{R}}\boldsymbol{\beta} \approx \binom{\boldsymbol{x}}{0}$
**Return:** $\boldsymbol{T}\boldsymbol{\beta}$
---

respect to a node feature $\boldsymbol{x}$, in that neighbors tend to have similar feature values, the raw feature is correlated positively with the propagated version $\boldsymbol{S}\boldsymbol{x}$ of the feature. By contrast, when a graph is heterophilous with respect to a feature, the correlation is negative. The least squares in ASGC is able to adapt to both cases and exploit this correlation, as well as correlations that occur when repeatedly propagating features (i.e., correlations of $\boldsymbol{x}$ with $\boldsymbol{S}^k\boldsymbol{x}$ for $k > 1$).

**Further remarks** In theory, as $K$ is raised to higher values, $\boldsymbol{T}$ will provide a sufficiently high-rank basis that $\boldsymbol{x}_{\text{ASGC}}$ will be arbitrarily close to $\boldsymbol{x}$, even if $R$ is very high. Then ASGC would have essentially the same performance as using the raw feature. While this issue could be resolved by introducing a smaller regularization term for the remaining coefficients, we find that this is generally not a problem over reasonable values of $K$ on real-world graphs; hence, for simplicity, we do not introduce this further regularization.

Pseudocode to filter a single feature is given in Algorithm 1. This algorithm is applied independently to all features; note that this is trivially parallelizable across features. After this, as in SGC, the filtered features are passed as input to a logistic regression classifier for node classification. The core computations in Algorithm 1 are 1) creation of the matrix $\boldsymbol{T}$ by multiplying $\boldsymbol{x}$ by $\boldsymbol{S}$ up to $K$ times, for which the time complexity is $O(mK)$, where $m$ is the number of edges; and 2) linear least squares with a matrix of dimensionality $(n + 1) \times (K + 1)$, for which the complexity is $O(nK^2)$.

**Spectral Interpretation of ASGC** ASGC admits an interesting alternate interpretation based on a spectral view of Equation 4. Let $\boldsymbol{S} = \boldsymbol{Q}\,\text{diag}(\boldsymbol{\lambda})\boldsymbol{Q}^\top$ be an eigendecomposition of $\boldsymbol{S}$, and let $\boldsymbol{\gamma} = \boldsymbol{Q}^{-1}\boldsymbol{x}$, that is, $\boldsymbol{\gamma}$ is the graph Fourier transform of the feature $\boldsymbol{x}$. The central objective in ASGC is the norm of the residual of the least squares in Equation 4. As in Parseval's Theorem, due to the orthogonality of $\boldsymbol{Q}$, this norm is invariant under the graph Fourier transform:

$$\|\boldsymbol{x}_{\text{ASGC}} - \boldsymbol{x}\|^2 = \|\boldsymbol{T}\boldsymbol{\beta} - \boldsymbol{x}\|^2 = \left\|\boldsymbol{Q}^\top\left(\boldsymbol{T}\boldsymbol{\beta} - \boldsymbol{x}\right)\right\|^2 = \left\|\boldsymbol{Q}^{-1}\left(\boldsymbol{T}\boldsymbol{\beta} - \boldsymbol{x}\right)\right\|^2.$$

Recall that each column of $\boldsymbol{T}$ is of the form $\boldsymbol{S}^i\boldsymbol{x}$ for some nonnegative power $i$. Then

$$\boldsymbol{Q}^{-1}\boldsymbol{S}^i\boldsymbol{x} = \boldsymbol{Q}^{-1}\left(\boldsymbol{Q}\,\text{diag}(\boldsymbol{\lambda})^i\boldsymbol{Q}^\top\right)(\boldsymbol{Q}\boldsymbol{\gamma}) = \text{diag}(\boldsymbol{\lambda})^i\boldsymbol{\gamma},$$

and the minimization objective can be written as

$$\|\boldsymbol{x}_{\text{ASGC}} - \boldsymbol{x}\|^2 = \left\|\boldsymbol{Q}^{-1}\boldsymbol{T}\boldsymbol{c} - \boldsymbol{Q}^{-1}\boldsymbol{x}\right\|^2 = \|\text{diag}(\boldsymbol{\gamma})\boldsymbol{V}_{\boldsymbol{\lambda}}\boldsymbol{\beta} - \boldsymbol{\gamma}\|^2 = \|\text{diag}(\boldsymbol{\gamma})\left(\boldsymbol{V}_{\boldsymbol{\lambda}}\boldsymbol{\beta} - \boldsymbol{1}\right)\|^2,$$

where, letting superscript $\circ i$ denote the entrywise $i^{\text{th}}$ power of a vector,

$$\boldsymbol{V}_{\boldsymbol{\lambda}} = \left(\boldsymbol{\lambda}^{\circ 0}; \boldsymbol{\lambda}^{\circ 1}; \boldsymbol{\lambda}^{\circ 2}; \ldots; \boldsymbol{\lambda}^{\circ K}\right)$$

is the Vandermonde matrix of powers $0$ to $K$ of the eigenvalues of $\boldsymbol{S}$. Note that multiplying $\boldsymbol{V}_{\boldsymbol{\lambda}}$ by the vector $\boldsymbol{\beta}$ yields the values of the polynomial with coefficients $\boldsymbol{\beta}$, evaluated at the eigenvalues $\boldsymbol{\lambda}$. Hence ASGC can be interpreted as fitting a $K$-degree polynomial over the graph's eigenvalues, with the target being all-ones. The value the polynomial takes over each eigenvalue represents how the

learned filter scales the component of the feature $x$ which is along the direction of the corresponding eigenvector of $S$; the all-ones target corresponds to a do-nothing filter. The least squares loss is weighed proportionately to the magnitude of this component at each eigenvalue. That $K$ is small, and the use of regularization on the zeroth coefficient, precludes the learned filter actually being the do-nothing filter, and instead results in a simple polynomial which adapts to the feature.

## 4  Motivating Example

We now use a synthetic network to demonstrate the capability of ASGC, and the potential deficiencies of SGC, at denoising a single heterophilous feature. We propose Featured SBMs, which augment stochastic block models (SBMs) (Holland et al., 1983) with a single feature; we note that our FSBMs can be seen as a simplified variant of recently studied Contextual SBMs (Deshpande et al., 2018).

**Definition 1** (Featured SBM). *An SBM graph $G$ has $n$ nodes partitioned into $r$ communities $C_1, C_2, \ldots, C_r$, with intra- and inter- community edge probabilities $p$ and $q$. Let $c_1, c_2, \ldots, c_r \in \{0,1\}^n$ be indicator vectors for membership in each community, i.e., the $j^{th}$ entry of $c_i$ is 1 if the $j^{th}$ node is in $C_i$ and 0 otherwise. A Featured SBM (FSBM) is such a graph model $G$, plus a feature vector $x = f + \eta \in \mathbb{R}^n$, where $\eta \sim \mathcal{N}(0, \sigma^2 I)$ is zero-centered, isotropic Gaussian noise and $f = \sum_i \mu_i c_i$ for some $\mu_1, \mu_2, \ldots, \mu_r \in \mathbb{R}$, which are the expected feature values of each community.*

We consider FSBMs with $n = 1000$, 2 equally-sized communities $C_+$ and $C_-$, feature means $\mu_+ = +1$, and $\mu_- = -1$, and noise variance $\sigma = 1$. Thus, there are 500 nodes in each community; calling these communities 'plus' and 'minus,' the feature mean is $+1$ for nodes in the former and $-1$ for the latter, to which standard normal noise is added. We generate different graphs by setting the expected degree of all nodes to 10 (that is, $\frac{1}{2}(p + q) \cdot n = 10$) , then varying the intra- and inter-community edge probabilities $p$ and $q$ from $p \gg q$ (highly homophilous, in that 'plus' nodes are much more likely to connect to other 'plus' nodes than to 'minus' nodes) to $q \gg p$ (highly heterophilous, in that 'plus' nodes tend to connect to 'minus' nodes). See Figure 1 for illustration.

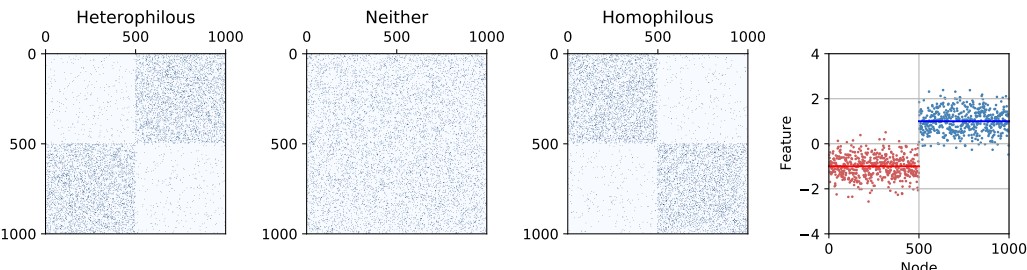

Figure 1: Synthetic dataset visualization. Left: 3 sample adjacency matrices, from the highly heterophilous ($q \ll p$) to the highly homophilous ($p \gg q$); for visual clarity, these graphs are 10 times denser the description in Section 4. Right: Feature values by node; note the feature means.

We seek to denoise the feature by exploiting the graph, which should result in the feature values moving towards the means of the respective communities. We employ SGC and our ASGC, both with number of hops $K = 2$. Figure 2 (left) shows the deviation from the feature means after denoising. It also shows the proportion of nodes whose filtered feature differs from the community mean in sign, that is, the error when classifying nodes into $C_+$ and $C_-$. By both metrics, ASGC outperforms SGC on heterophilous graphs, while SGC outperforms ASGC on homophilous graphs. Both methods can lose accuracy relative to just using the raw feature when the graph is neither homophilous nor heterophilous, that is, when the graph is not informative about the communities. However, the performance of ASGC increases similarly as the graph becomes either more heterophilous or homophilous, whereas SGC's performance improves significantly less in the heterophilous direction. Finally, the performance gap between the two is smaller on homophilous graphs, particularly on sign accuracy, suggesting that ASGC can better adapt to varying degrees of homophily/heterophily. We examine the highly heterophilous case in more detail in Figure 2 (right), which shows the distributions of the feature before and after filtering. The fixed propagation of SGC tends towards merging the two communities' feature distributions; by contrast, ASGC is able to keep them separated, preserves the feature means, and pulls the feature distributions towards the respective community means.

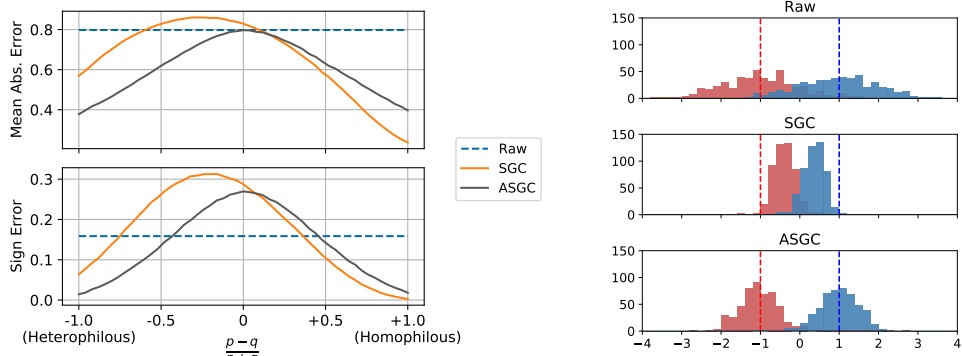

Figure 2: Left: Denoising results on the synthetic graphs using SGC and ASGC with number of hops $K = 2$. 'Raw' shows the error when no filtering method is applied. ASGC and SGC are more effective at denoising on heterophilous and homophilous networks, respectively, and ASGC is more effective overall. Right: Distribution of the feature values before and after applying each of the filtering methods on a very heterophilous synthetic graph ($\frac{p-q}{p+q} = -9/10$), separated by ground-truth communities. SGC tends to merge the two communities, while ASGC is able to keep them separated.

## 5   Theoretical Guarantees

To support our empirical investigation in Section 4, we now theoretically verify the limitations and capabilities of SGC and ASGC at denoising FSBM networks. For simplicity, we analyze SGC without the addition of a self-loop (that is, using $S$ in Equation 1 rather than $\tilde{S}$); the distinction between the two in the analysis vanishes as the number of intra-community edges grows, i.e., if $n \cdot p$ is high. Further, we assume that the regularization hyperparameter $R$ for ASGC is high, in which case the coefficient $\beta_0$ is fixed to zero, or equivalently, the column $S^0 x$ is removed from the Krylov matrix $T$ in Equations 3 and 4. Finally, we analyze using expected adjacency matrices from the model, though we conjecture that via concentration bounds one could extend the following results to the sampled setting (Spielman, 2012). Here, we analyze FSBMs with 2 equally-sized communities. In Appendix 9.1, we prove that these results extend to an arbitrary number of communities. Unless otherwise specified, we use the same notation as Definition 1.

**Theorem 1** (Effect of SGC on FSBM Networks). *Consider FSBMs having 2 equally-sized communities with indicator vectors $c_u$ and $c_v$, expected adjacency matrix $A$, and feature vector $x = f + \eta$. Let $x_{SGC}$ be the feature vector returned by applying SGC, with number of hops $K$, to $A$ and $x$. Further, let $\bar{\mu} = \frac{1}{2}(\mu_u + \mu_v)$ be the average of the feature means. Then, $x_{SGC} = f' + \theta_u c_u + \theta_v c_v$, where $f' = \lambda_2^K f + (1 - \lambda_2^K)(\bar{\mu}\mathbf{1})$, and $\theta_u$ and $\theta_v$ are both distributed according to $\mathcal{N}\left(0, \frac{1}{n}(1 + \lambda_2^{2K})\sigma^2\right)$.*

*Proof.* In expectation, an entry $A_{ij}$ of the adjacency matrix of the graph is $p$ if both $i, j \in C_u$ or both $i, j \in C_v$, and it is $q$ otherwise. The eigendecomposition $Q \operatorname{diag}(\lambda) Q^\top$ of the associated normalized adjacency matrix $S$ has two nonzero eigenvalues: $\lambda_1 = 1$, with eigenvector $q_1 = (1/\sqrt{n})\mathbf{1} = (1/\sqrt{n})(c_u + c_v)$, and $\lambda_2 = \frac{p-q}{p+q}$, with $q_2 = (1/\sqrt{n})(c_u - c_v)$. In the following analysis, we use the fact that zero-centered, isotropic Gaussian distributions are invariant to rotation, meaning $Q^\top \eta = \eta' \sim \mathcal{N}(0, \sigma^2 I)$ for any orthonormal matrix $Q$.

$$
\begin{aligned}
x_{\text{SGC}} = S^K x &= Q \operatorname{diag}(\lambda)^K Q^\top (\mu_u c_u + \mu_v c_v + \eta) \\
&= q_1 q_1^\top (\mu_u c_u + \mu_v c_v + \eta) + \lambda_2^K q_2 q_2^\top (\mu_u c_u + \mu_v c_v + \eta) \\
&= q_1 \left( \frac{\sqrt{n}}{2}(\mu_u + \mu_v) + \eta_1' \right) + \lambda_2^K q_2 \left( \frac{\sqrt{n}}{2}(\mu_u - \mu_v) + \eta_2' \right) \\
&= \left( \frac{1}{2}(\mu_u + \mu_v) + \frac{1}{\sqrt{n}}\eta_1' \right)(c_u + c_v) + \lambda_2^K \left( \frac{1}{2}(\mu_u - \mu_v) + \frac{1}{\sqrt{n}}\eta_2' \right)(c_u - c_v) \\
&= \quad \lambda_2^K (\mu_u c_u + \mu_v c_v) + (1 - \lambda_2^K) \cdot \frac{1}{2}(\mu_u + \mu_v)(c_u + c_v) \\
&\quad + \frac{1}{\sqrt{n}}\left( \eta_1' + \lambda_2^K \eta_2' \right) c_u + \frac{1}{\sqrt{n}}\left( \eta_1' - \lambda_2^K \eta_2' \right) c_v \\
&= \lambda_2^K f + (1 - \lambda_2^K)(\bar{\mu}\mathbf{1}) + \theta_+ c_u + \theta_- c_v,
\end{aligned}
$$

where $\theta_\pm = \frac{1}{\sqrt{n}}(\eta'_1 \pm \lambda_2^K \eta'_2)$, which has the specified distribution. $\qquad\square$

Note that $\lambda_2 = \frac{p-q}{p+q} \in [-1, +1]$, with negative values indicating heterophily ($p < q$) and positive values indicating homophily ($p > q$). SGC only preserves the feature means in certain limiting cases. In particular, this occurs as $\lambda_2 \to +1$, or as $\lambda_2 \to -1$ if $K$ is even; then $\lambda_2^K \to 1$, so the expected filtered feature vector $\boldsymbol{f}' \to \boldsymbol{f}$. On the other hand, if $\lambda_2 \to 0$, then $\lambda_2^K \to 0$ and $\boldsymbol{f}' \to \bar{\mu}\mathbf{1}$, that is, the feature value means are averaged between the communities. Finally, if $\lambda_2 \to -1$ and $K$ is odd, then $\lambda_2^K \to -1$ and $\boldsymbol{f}' = \bar{\mu}\mathbf{1} + (\bar{\mu}\mathbf{1} - \boldsymbol{f}) = \mu_v \boldsymbol{c_u} + \mu_u \boldsymbol{c_v}$: the feature value means are entirely exchanged across the communities. By contrast, ASGC preserves the means, while similarly reducing noise by an $O(n)$ factor, with much looser restrictions on $\lambda_2$ and $K$:

**Theorem 2** (Effect of ASGC on FSBM Networks). *Consider FSBMs with $p \neq q$ having 2 equally-sized communities with community indicator vectors $\boldsymbol{c_u}$ and $\boldsymbol{c_v}$, expected adjacency matrix $\boldsymbol{A}$, and feature vector $\boldsymbol{x} = \boldsymbol{f} + \boldsymbol{\eta}$. Let $\boldsymbol{x}_{ASGC}$ be the feature vector returned by applying ASGC, with number of hops $K \geq 2$, to $\boldsymbol{A}$ and $\boldsymbol{x}$. Then $\boldsymbol{x}_{ASGC} = \boldsymbol{f} + \theta'_+ \boldsymbol{c_u} + \theta'_- \boldsymbol{c_v}$, where $\theta'_+$ and $\theta'_-$ are both distributed according to $\mathcal{N}\left(0, \frac{2}{n}\sigma^2\right)$.*

*Proof.* The least squares in ASGC is equivalent to projecting the feature $\boldsymbol{x}$ onto the column span of the Krylov matrix $\boldsymbol{T} = \left(\boldsymbol{S}^1 \boldsymbol{x}; \boldsymbol{S}^2 \boldsymbol{x}; \dots; \boldsymbol{S}^K \boldsymbol{x}\right)$. Observe that the column span of $\boldsymbol{T}$ is contained in the column span of $\boldsymbol{S}$. Further, $\boldsymbol{S}$ is rank-2 (by the assumption that $p \neq q$), so with probability 1 over the distribution of $\boldsymbol{\eta}$, as long as $K \geq 2$, the column span of $\boldsymbol{T}$ equals that of $\boldsymbol{S}$. Thus ASGC projects $\boldsymbol{x}$ onto the column span of $\boldsymbol{S}$, i.e., the span of $\boldsymbol{q_1}, \boldsymbol{q_2}$, the eigenvectors of $\boldsymbol{S}$. The following analysis proceeds exactly like the one for SGC, just without the terms for the eigenvalue $\lambda_2$, so we use the same notation and abbreviate the steps:

$$\begin{aligned}
\boldsymbol{x}_{\text{ASGC}} &= \boldsymbol{Q}\boldsymbol{Q}^\top \boldsymbol{x} \\
&= \boldsymbol{q_1}\boldsymbol{q_1}^\top (\mu_u \boldsymbol{c_u} + \mu_v \boldsymbol{c_v} + \boldsymbol{\eta}) + \boldsymbol{q_2}\boldsymbol{q_2}^\top (\mu_u \boldsymbol{c_u} + \mu_v \boldsymbol{c_v} + \boldsymbol{\eta}) \\
&= \boldsymbol{f} + \theta'_+ \boldsymbol{c_u} + \theta'_- \boldsymbol{c_v},
\end{aligned}$$

where $\theta'_\pm = \frac{1}{\sqrt{n}}(\eta'_1 \pm \eta'_2)$, which has the specified distribution. $\qquad\square$

Observe that in the sampled setting, standard matrix concentration results can be used to show that, while $\boldsymbol{S}$ will be full rank with high probability, it will have two outlying eigenvalues, corresponding to eigenvectors close to $\boldsymbol{q_1}$ and $\boldsymbol{q_2}$ (Spielman, 2012). It is well known that the span of the Krylov matrix $\boldsymbol{T}$ will align well with these outlying eigendirections (Saad, 2011). Thus, we expect the projection $\boldsymbol{Q}\boldsymbol{Q}^\top \boldsymbol{x} = \boldsymbol{Q}\boldsymbol{Q}^\top \boldsymbol{f} + \boldsymbol{Q}\boldsymbol{Q}^\top \boldsymbol{\eta}$ to still approximately preserve $\boldsymbol{f}$. At the same time, $\boldsymbol{Q}\boldsymbol{Q}^T \boldsymbol{\eta}$ is the projection of a random Gaussian vector $\boldsymbol{\eta}$ onto a fixed $K$-dimensional subspace. Thus, we will have $\left\|\boldsymbol{Q}\boldsymbol{Q}^\top \boldsymbol{\eta}\right\|_2^2 \approx \frac{K}{n} \left\|\boldsymbol{\eta}\right\|_2^2$, so ASGC will still perform significant denoising when $K$ is small.

## 6  Related Work

**Deep graph models**    As discussed previously, the SGC algorithm is a drastic simplification of the graph convolutional network (GCN) model (Kipf & Welling, 2017). GCNs learn a sequence of node representations that evolve via repeated propagation through the graph and nonlinear transformations. The starting node representations $\boldsymbol{H}^{(0)}$ are set to the input feature matrix $\boldsymbol{X} \in \mathbb{R}^{n \times f}$, where $f$ is the number of features. The $k^{\text{th}}$-step representations are $\boldsymbol{H}^{(k)} = \sigma\left(\boldsymbol{S}\boldsymbol{H}^{(k-1)}\boldsymbol{\Theta}^{(k)}\right)$, where $\boldsymbol{\Theta}^{(k)}$ is the learned linear transformation matrix for the $k^{\text{th}}$ layer and $\sigma$ is a nonlinearity like ReLU. After $K$ such steps, the representations are used to classify the nodes via a softmax layer, and the whole model is trained end-to-end via gradient descent. Wu et al. (2019) observe that if the nonlinearities are ignored, all of the linear transformations collapse into a single one, while the repeated multiplications by $\boldsymbol{S}$ collapse into a single one by $\boldsymbol{S}^K$; this yields their algorithm of the SGC filter (Equation 1) followed by logistic regression. GCNs have spawned streamlined versions like FastGCN (Chen et al., 2018), as well as more complicated variants like graph isomorphism networks (GINs) (Xu et al., 2019) and graph attention networks (GATs) (Veličković et al., 2018); despite being much simpler and faster than these competitors, SGC manages similar performance on common benchmarks, though, based on the analysis of NT & Maehara (2019), this may be due in part to the simplicity of the benchmark datasets in that they mainly exhibit homophily/assortativity.

Table 1: Statistics of datasets used in our experiments, separated by homophilous vs heterophilous.

| Dataset | CORA | CITE. | PUBM. | COMP. | PHOTO | CHAM. | SQUI. | ACTOR | TEXAS | CORN. |
|---|---|---|---|---|---|---|---|---|---|---|
| Nodes | 2708 | 3327 | 19717 | 13752 | 7650 | 2277 | 5201 | 7600 | 183 | 183 |
| Edges | 5278 | 4552 | 44324 | 245861 | 119081 | 31421 | 198493 | 26752 | 295 | 280 |
| Features | 1433 | 3703 | 500 | 767 | 745 | 2325 | 2089 | 932 | 1703 | 1703 |
| Classes | 7 | 6 | 3 | 10 | 8 | 5 | 5 | 5 | 5 | 5 |
| $H(\mathcal{G})$ | 0.825 | 0.718 | 0.792 | 0.802 | 0.849 | 0.247 | 0.217 | 0.215 | 0.057 | 0.301 |

**Addressing heterophily** Like our work, some other recent methods attempt to address node heterophily. Gatterbauer (2014) and Zhu et al. (2020a) augment classical feature propagation and GCNs, respectively, to accommodate heterophily by modifying feature propagation based on node classes. Zhu et al. (2020b) and Yan et al. (2021) analyze common structures in heterophilous graphs and the failure points of GCNs, then propose GCN variants based on their analyses. The Geom-GCN paper of Pei et al. (2020) introduces several of the real-world heterophilous networks which are commonly used in related papers, including this one. Their method allows for long-range feature propagation based on similarity of pre-trained node embeddings. The preceding is just a sample of recent works in this area, which has seen a surge of activity (Liu et al., 2020; Luan et al., 2021; Suresh et al., 2021). We note that, like the GNNs of Kipf & Welling (2017) but unlike SGC and our ASGC, almost all of these methods are based on deep learning and are trained via backpropagation through repeated feature propagation and linear transformation steps, and hence incur an associated speed and memory requirement. Understanding and implementing these methods is also more complicated relative to our method, which just constitutes a learned feature filter and logistic regression. We mainly compare our results with the deep method which is most similar to ours, Generalized PageRank GNN (GPR-GNN) (Chien et al., 2021). Like ASGC, GPR-GNN produces node representations by linear combination of propagated versions of node features; unlike ASGC, the raw features are first transformed by a neural network, and parameters for this network, as well as the linear combination coefficients, are learned by backpropagation using the known node labels. To our knowledge, our work is the first to show that heterophily can be handled using just feature pre-processing.

## 7 Empirical Performance

We test the performance of ASGC for the node classification task on a benchmark of real-world datasets given in Table 1, and compare with SGC and several deep methods.

**Real-world datasets** We experiment on 10 commonly-used datasets, the same collection of datasets as Chien et al. (2021). CORA, CITESEER, and PUBMED are citation networks which are common benchmarks for node classification (Sen et al., 2008; Namata et al., 2012); these have been used for evaluation on the GCN (Kipf & Welling, 2017) and GAT (Veličković et al., 2018) papers, in addition to SGC itself. The features are bag-of-words representations of papers, and the node labels give the topics of the paper. COMPUTERS and PHOTO are segments of the Amazon co-purchase graph (McAuley et al., 2015; Shchur et al., 2018); features are derived from product reviews, and labels are product categories. These first 5 datasets are considered assortative/homophilous; the remaining 5 datasets, which are disassortative/heterophilous, come from the Geom-GCN paper (Pei et al., 2020), which also introduces the following measure of of a network's homophily: $H(\mathcal{G}) = \frac{1}{|V|} \sum_{v \in V} \frac{\text{\# } v\text{'s neighbors with the same label as } v}{\text{\# neighbors of } v} \in [0, 1]$. We include this statistic in Table 1. The latter 5 datasets have much lower values of $H(\mathcal{G})$. CHAMELEON and SQUIRREL are hyperlink networks of pages in Wikipedia which concern the two topics (Rozemberczki et al., 2021). Features derive from text in the pages, and labels correspond to the amount of web traffic to the page, split into 5 categories. ACTOR is the actor-only induced subgraph of the film-director-actor-writer network of Tang & Liu (2009). Nodes and edges represent actors and co-occurrence on a Wikipedia page. Features are based on keywords on the webpage, and labels derive from the number of words on the page, split into 5 categories. Finally, TEXAS and CORNELL are hyperlink networks from university websites (Craven et al., 2000); features derive from webpage text, and the labels represent the type of page: student, project, course, staff, or faculty.

**Implementation** The SGC and ASGC algorithms are implemented in Python using NumPy (Harris et al., 2020) for least squares regression and other linear algebraic computations. We use scikit-

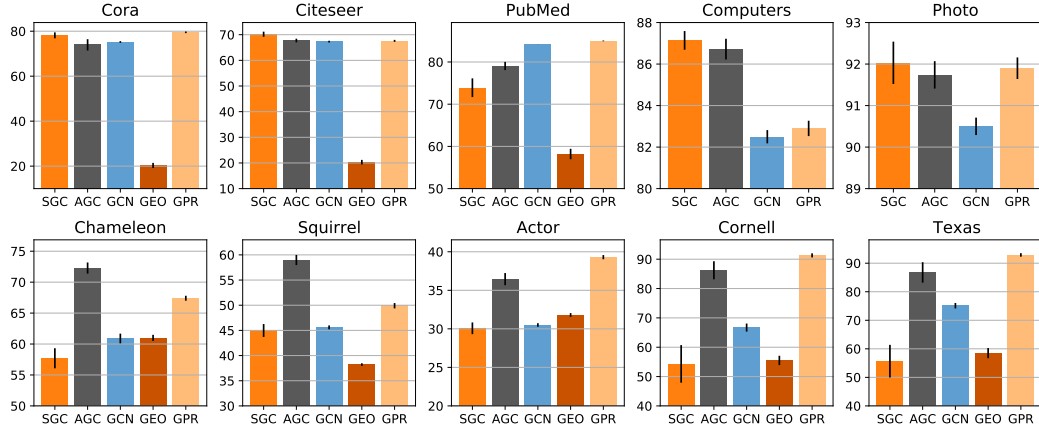

Figure 3: Test classification accuracy on the benchmark of datasets from Table 1 for selected methods: 2 non-deep, SGC and ASGC; and 3 deep, GCN, Geom-GCN, and GPR-GNN. Error bars show the 95% confidence intervals. SGC is generally competitive with the deep methods on the homophilous datasets (top row), but not so on the heterophilous ones, whereas ASGC is competitive throughout.

learn (Pedregosa et al., 2011) for logistic regression with 1,000 maximum iterations and otherwise default settings. For our implementations of SGC and ASGC, we treat each network as undirected, in that if edge $(i, j)$ appears, we also include edge $(j, i)$. Like Chien et al. (2021), we use random 60%/20%/20% splits as training/validation/test data for the 5 heterophilous datasets, as in Pei et al. (2020), and use random 2.5%/2.5%/95% splits for the homophilous datasets, which is closer to the original setting from Kipf & Welling (2017) and Shchur et al. (2018). We release code in the form of a Jupyter notebook (Pérez & Granger, 2007) demo which is available at github.com/schariya/adaptive-simple-convolution.

**Hyperparameter settings**   We tune the number of hops over $K \in \{1, 2, 4, 8\}$, roughly covering the range analyzed in Wu et al. (2019), and the regularization strength $R = \sqrt{n \cdot R'}$ over $\log_{10}(R') \in \{-4, -3, -2, -1, 0\}$. This dependency on the number of nodes $n$ allows the regularization loss to scale with the least squares loss, which generally grows linearly with $n$. In Appendix 9.2, we report results for some additional experiments investigating the effect of fixing these hyperparameters.

**Classification results**   We apply our implementations of SGC and ASGC to these datasets and report the mean test accuracy across 10 random splits of the data. As a baseline, we also report the accuracy of logistic regression on the 'raw,' unfiltered features, ignoring the graph. We compare to results from Chien et al. (2021) for 9 deep methods applied to these datasets. These methods are 1) a multi-layer perceptron which ignores the graph; 2) GCN; 3) GAT; 4) SAGE (Hamilton et al., 2017); 5) JKNet (Xu et al., 2018); 6) GCN-Cheby (Defferrard et al., 2016); 7) Geom-GCN; 8) APPNP (Klicpera et al., 2019); and 9) GPR-GNN. Full results are given in Table 2 in Appendix 9.3.

We plot accuracies for selected methods in Figure 3. In addition to SGC and ASGC, we include 3 deep methods: 'vanilla' GCN; Geom-GCN, which originated the heterophilous datasets; and GPR-GNN, a recent method claiming state-of-the-art performance. We find that SGC is generally competitive with the deep methods on the homophilous datasets, but not so on the heterophilous ones, whereas ASGC is generally competitive throughout. Interestingly, the datasets on which GPRGNN significantly outperforms ASGC (PUBMED, ACTOR, TEXAS, CORNELL) are exactly those on which a multi-layer perceptron significantly outperforms logistic regression; note that the latter two methods both ignore the graph. This suggests that the some nonlinear processing of the node features may be key to performance on these networks, separate from how the graph is exploited. To compactly compare all 12 of the methods across these 10 datasets, we aggregate the performance across the datasets as follows. For each dataset, we calculate the accuracy of each method as a proportion of the accuracy of the best method. We plot the mean and the minimum across the 10 datasets of each method's proportional accuracies. See Figure 4. GPRGNN and ASGC achieve the highest mean

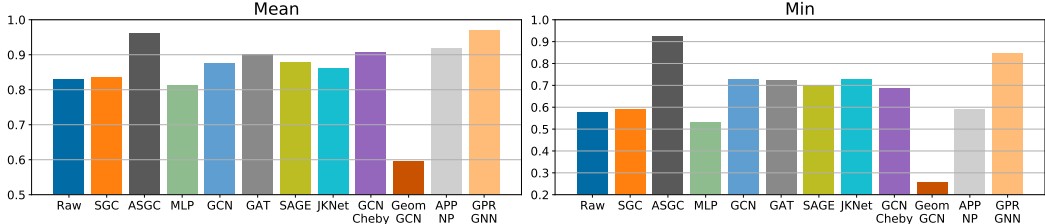

Figure 4: Test accuracy as a proportion of the best method's accuracy; mean and minimum performance over the 10 networks. The 3 non-deep (logistic regression) methods are on the left. ASGC achieves the highest minimum performance, and is competitive with GPRGNN on the mean.

performance. Further, ASGC achieves the highest minimum performance: at worst, it achieves over 90% of the best method's test accuracy on each of the datasets.

## 8    Conclusion, Limitations, and Broader Impact

Building on SGC, we propose a feature filtering technique, ASGC, based on feature propagation and least-squares regression. We propose a natural class of synthetic featured networks, FSBMs, and show both empirically and with theoretical guarantees that ASGC can denoise both homophilous and heterophilous FSBMs, whereas SGC is inappropriate for the latter. Further, we find that ASGC is generally competitive with recent deep learning-based methods on a benchmark of real-world datasets, covering both homophilous and heterophilous networks. Our results suggest that deep learning is not strictly necessary for handling heterophily, and that even a simple feature pre-processing method can be competitive. We hope that, like SGC, ASGC can serve as a good first method to try, especially for node classification on heterophilous networks, and a baseline for future works.

We note certain limitations of our work. First, our theoretical guarantees are in the fairly idealized setting of FSBM networks; we provide no general guarantee of performance. Second, our empirical evaluations on real-world datasets do not include any truly large-scale datasets ($n \sim 10^6$ or higher), though our collection of 10 datasets is a fairly broad selection amongst those which are commonly used in related papers about heterophily. Finally, as we note in Section 6, there are many more recent works and methods for handling heterophily to which we could compare our results. We limit ourselves to a subset due to the amount of activity in this area; further, we are not asserting absolute state-of-the-art results in this area, merely that a simple method like ours can be competitive.

We do not anticipate direct negative societal impacts of our work, but improved methods for network analysis, including those for node classification, have social consequences. For example, improved graph-based recommendations may be associated with negative impacts to well-being due to increased social media use (Verduyn et al., 2017; Kelly et al., 2018). They might also contribute to concerns of filter bubbles and polarization (Nguyen et al., 2014; Lee et al., 2014; Musco et al., 2018).

### Acknowledgments and Disclosure of Funding

This project was partially supported by an Adobe Research grant, a Google Research Scholar Award, and NSF Grants 2046235 and 1763618. We also thank Raj Kumar Maity and Konstantinos Sotiropoulos for helpful conversations.

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
