$\boldsymbol{S}$; this polynomial is set so that $\boldsymbol{x}_{\text{ASGC}} \approx \boldsymbol{x}$:

$$\boldsymbol{x}_{\text{ASGC}} = \left(\sum\nolimits_{k=0}^{K} \beta_k \boldsymbol{S}^k\right) \boldsymbol{x} \approx \boldsymbol{x}, \tag{2}$$

where $K$ is a hyperparameter which, as in SGC, controls the radius of feature propagation, and the coefficients $\boldsymbol{\beta} \in \mathbb{R}^{K+1}$ are learned by minimizing the approximation error in a least squares sense. Note that, unlike SGC, we do not add self-loops to $\boldsymbol{S}$. This approximation error would be trivially minimized with $\beta_0 = 1$ and all other coefficients set to zero, resulting in $\boldsymbol{x}_{\text{ASGC}} = \boldsymbol{S}^0 \boldsymbol{x} = \boldsymbol{x}$, so we regularize the magnitude of $\beta_0$.

More concretely, let $\boldsymbol{T} \in \mathbb{R}^{n \times (K+1)}$ denote the Krylov matrix generated by multiplying a feature vector $\boldsymbol{x} \in \mathbb{R}^n$ by the normalized adjacency matrix $\boldsymbol{S}$ up to some $K \in \mathbb{Z}_{>0}$ times:

$$\boldsymbol{T} = \left(\boldsymbol{S}^0 \boldsymbol{x}; \boldsymbol{S}^1 \boldsymbol{x}; \boldsymbol{S}^2 \boldsymbol{x}; \ldots; \boldsymbol{S}^K \boldsymbol{x}\right). \tag{3}$$

Here the leftmost column of $\boldsymbol{T}$ is just the raw feature $\boldsymbol{x}$, and each column represents the feature generated by propagating the feature vector to its left across the graph, i.e., by multiplying once by $\boldsymbol{S}$. We produce a filtered version $\boldsymbol{x}_{\text{ASGC}}$ of the raw feature $\boldsymbol{x}$ by linear combination of the columns of $\boldsymbol{T}$. That is, $\boldsymbol{x}_{\text{ASGC}} = \boldsymbol{T}\boldsymbol{\beta}$, and we set the combination coefficients $\boldsymbol{\beta} \in \mathbb{R}^{k+1}$ by minimizing a loss function as follows:

$$\min_{\boldsymbol{\beta}} \left(\|\boldsymbol{T}\boldsymbol{\beta} - \boldsymbol{x}\|_2^2 + (R\beta_0)^2\right). \tag{4}$$

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

 $T$ equals that of $S$. Thus ASGC projects $x$ onto the column span of $S$, i.e., the span of $q_1, q_2$, the eigenvectors of $S$. The following analysis proceeds exactly like the one for SGC, just without the terms for the eigenvalue $\lambda_2$, so we use the same notation and abbreviate the steps:

$$
\begin{aligned}
x_{\text{ASGC}} &= QQ^\top x \\
&= q_1 q_1^\top (\mu_u c_u + \mu_v c_v + \eta) + q_2 q_2^\top (\mu_u c_u + \mu_v c_v + \eta) \\
&= f + \theta'_+ c_u + \theta'_- c_v,
\end{aligned}
$$

where $\theta'_{\pm} = \frac{1}{\sqrt{n}}(\eta'_1 \pm \eta'_2)$, which has the specified distribution. $\qquad\square$

Observe that in the sampled setting, standard matrix concentration results can be used to show that, while $S$ will be full rank with high probability, it will have two outlying eigenvalues, corresponding to eigenvectors close to $q_1$ and $q_2$ (Spielman, 2012). It is well known that the span of the Krylov matrix $T$ will align well with these outlying eigendirections (Saad, 2011). Thus, we expect the projection $QQ^\top x = QQ^\top f + QQ^\top \eta$ to still approximately preserve $f$. At the same time, $QQ^T \

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

# 9 Appendix

## 9.1 Multi-Community FSBM Proofs

We now prove generalizations of the theorems from Section 5 for FSBMs that may have more than 2 communities, i.e., with $r \geq 2$ from Definition 1, and otherwise the same assumptions. The theorem statements and their implications are essentially the same. The proofs are also similar in concept, just using a different projection matrix $Q$, though the proof for the generalized Theorem 1 is significantly more involved.

**Theorem 1** (Effect of SGC on FSBM Networks). *Consider FSBMs having $r$ equally-sized communities with indicator vectors $c_1, c_2, \ldots, c_k$, expected adjacency matrix $A$, and feature vector $x = f + \eta$. Let $x_{SGC}$ be the feature vector returned by applying SGC, with number of hops $K$, to $A$ and $x$. Further, let $\bar{\mu} = \frac{1}{r} \sum_i \mu_i$ be the average of the feature means. Then, $x_{SGC} = f' + \sum_i \theta_i c_i$, where $f' = \lambda_2^K f + (1 - \lambda_2^K)(\bar{\mu} 1)$, and each $\theta_i$ is distributed according to $\mathcal{N} \left( 0, \frac{1}{n} \left( 1 + \lambda_2^{2K}(r-1) \right) \sigma^2 \right)$.*

*Proof.* Let $\hat{1} = (1/\sqrt{n}) 1$, where $1$ is the $n$-length all-ones vector. Let $C = (c_1; c_2; \ldots; c_r)$ and $Q = (\sqrt{r/n}) C$. Note that $\hat{1}$ has norm 1, and the columns of $Q$ are orthonormal. Finally, let $\lambda_2 = \frac{p-q}{p+(r-1)q}$. We we will not make use of this fact, but, as in the 2-community case from Section 5, this is still the second largest eigenvalue of $S$, and it now has multiplicity $r - 1$.

The expected adjacency matrix of the graph is

$$A = (p-q)CC^\top + q 1 1^\top = (p-q)(n/r)QQ^\top + qn\hat{1}\hat{1}^\top,$$

and the expected degree vector is

$$d = A1 = (p-q)(n/r)1 + qn1 = (p + (r-1)q) (n/r) 1,$$

yielding the expected normalized adjacency matrix

$$\begin{aligned}
S &= \frac{(p-q)(n/r)QQ^\top + qn\hat{1}\hat{1}^\top}{(p + (r-1)q)(n/r)} \\
&= \lambda_2 QQ^\top + \frac{qr}{p+(r-1)q}\hat{1}\hat{1}^\top \\
&= \lambda_2 QQ^\top + (1-\lambda_2)\hat{1}\hat{1}^\top \\
&= (QQ^\top)\left(\lambda_2 I + (1-\lambda_2)\hat{1}\hat{1}^\top\right).
\end{aligned}$$

Note that $(QQ^\top)^2 = QQ^\top$ and $(\hat{1}\hat{1}^\top)^2 = \hat{1}\hat{1}^\top$ since these are projection matrices. We show that

$$\left(\lambda_2 I + (1-\lambda_2)\hat{1}\hat{1}^\top\right)^K = \lambda_2^K I + \left(1 - \lambda_2^K\right)\hat{1}\hat{1}^\top$$

by induction as follows:

$$\begin{aligned}
\left(\lambda_2 I + (1-\lambda_2)\hat{1}\hat{1}^\top\right)^K &= \left(\lambda_2 I + (1-\lambda_2)\hat{1}\hat{1}^\top\right)\left(\lambda_2 I + (1-\lambda_2)\hat{1}\hat{1}^\top\right)^{K-1} \\
&= \left(\lambda_2 I + (1-\lambda_2)\hat{1}\hat{1}^\top\right)\left(\lambda_2^{K-1} I + \left(1 - \lambda_2^{K-1}\right)\hat{1}\hat{1}^\top\right) \\
&= \lambda_2^K I + \left(\lambda_2 \left(1 - \lambda_2^{K-1}\right) + (1-\lambda_2)\lambda_2^{K-1} + (1-\lambda_2)\left(1 - \lambda_2^{K-1}\right)\right)\hat{1}\hat{1}^\top \\
&= \lambda_2^K I + \left(1 - \lambda_2^K\right)\hat{1}\hat{1}^\top.
\end{aligned}$$

Using this result and the fact that $(QQ^\top)(\hat{1}\hat{1}^\top) = (\hat{1}\hat{1}^\top)(QQ^\top) = \hat{1}\hat{1}^\top$, we have

$$\begin{aligned}
S^K &= \left((QQ^\top)\left(\lambda_2 I + (1-\lambda_2)\hat{1}\hat{1}^\top\right)\right)^K \\
&= (QQ^\top)^K \left(\lambda_2 I + (1-\lambda_2)\hat{1}\hat{1}^\top\right)^K \\
&= (QQ^\top)\left(\lambda_2^K I + \left(1 - \lambda_2^K\right)\hat{1}\hat{1}^\top\right). \tag{5}
\end{aligned}$$

Now, as in the 2-community case, $\boldsymbol{Q}^\top \boldsymbol{\eta} = \boldsymbol{\eta}' \sim \mathcal{N}(0, \sigma^2 \boldsymbol{I})$, yielding

$$
\begin{aligned}
\boldsymbol{Q}^\top \boldsymbol{x} &= \boldsymbol{Q}^\top \left( \sum_i \mu_i \boldsymbol{c_i} + \boldsymbol{\eta} \right) \\
&= \left( \sqrt{r/n} \right) (\mu_1, \mu_2, \ldots, \mu_r)^\top + (\eta_1', \eta_2', \ldots, \eta_r')^\top, \\
\boldsymbol{Q}\boldsymbol{Q}^\top \boldsymbol{x} &= \sum_i \left( \mu_i + \left( \sqrt{r/n} \right)\eta_i' \right) \boldsymbol{c_i}, \text{ and} \\
(\hat{\boldsymbol{1}}\hat{\boldsymbol{1}}^\top)(\boldsymbol{Q}\boldsymbol{Q}^\top)\boldsymbol{x} &= \left( \tfrac{1}{r} \sum_i \left( \mu_i + \left( \sqrt{r/n} \right)\eta_i' \right) \right) \boldsymbol{1}.
\end{aligned} \tag{6}
$$

Finally, combining these equations with the expression for $\boldsymbol{S}^K$, we have

$$
\begin{aligned}
\boldsymbol{x}_{\text{SGC}} = \boldsymbol{S}^K \boldsymbol{x} &= \lambda_2^K \sum_i \left( \mu_i + \left( \sqrt{r/n} \right)\eta_i' \right) \boldsymbol{c_i} + (1 - \lambda_2^K) \left( \tfrac{1}{r} \sum_j \left( \mu_j + \left( \sqrt{r/n} \right)\eta_j' \right) \right) \boldsymbol{1} \\
&= \sum_i \left( \lambda_2^K \left( \mu_i + \left( \sqrt{r/n} \right)\eta_i' \right) + (1 - \lambda_2^K) \cdot \tfrac{1}{r} \sum_j \left( \mu_j + \left( \sqrt{r/n} \right)\eta_j' \right) \right) \boldsymbol{c_i} \\
&= \sum_i \left( \lambda_2^K \mu_i + (1 - \lambda_2^K) \bar{\mu} + \lambda_2^K \left( \sqrt{r/n} \right)\eta_i' + (1 - \lambda_2^K) \cdot \tfrac{1}{r} \sum_j \left( \sqrt{r/n} \right)\eta_j' \right) \boldsymbol{c_i} \\
&= \boldsymbol{f}' + \sum_i \left( \lambda_2^K \left( \sqrt{r/n} \right)\eta_i' + (1 - \lambda_2^K) \cdot \tfrac{1}{r} \sum_j \left( \sqrt{r/n} \right)\eta_j' \right) \boldsymbol{c_i},
\end{aligned}
$$

so the expectation $\boldsymbol{f}'$ of the filtered feature is as desired. Further, letting the noise term summands be $\theta_i \boldsymbol{c_i}$, we have

$$
\begin{aligned}
\theta_i &= \lambda_2^K \left( \sqrt{r/n} \right)\eta_i' + (1 - \lambda_2^K) \cdot \tfrac{1}{r} \sum_j \left( \left( \sqrt{r/n} \right)\eta_j' \right) \\
&= \left( \lambda_2^K + \tfrac{1}{r}(1 - \lambda_2^K) \right) \left( \sqrt{r/n} \right)\eta_i' + (1 - \lambda_2^K) \left( \tfrac{1}{r} \sum_{j \neq i} \left( \sqrt{r/n} \right)\eta_j' \right),
\end{aligned}
$$

which is normally distributed with mean 0 and variance

$$
\begin{aligned}
&\left( \lambda_2^K + \tfrac{1}{r}(1 - \lambda_2^K) \right)^2 \cdot \tfrac{r}{n}\sigma^2 + \left( 1 - \lambda_2^K \right)^2 \cdot \tfrac{1}{r^2}(r-1) \cdot \tfrac{r}{n}\sigma^2 \\
&= \tfrac{\sigma^2}{n} \left( \left( r\lambda_2^K + (1 - \lambda_2^K) \right)^2 \cdot \tfrac{1}{r} + (1 - \lambda_2^K)^2 \left( 1 - \tfrac{1}{r} \right) \right) \\
&= \tfrac{\sigma^2}{n} \left( 1 + \lambda_2^{2K}(r-1) \right),
\end{aligned}
$$

so the noise variance is also as desired. $\square$

**Theorem 2** (Effect of ASGC on FSBM Networks). *Consider FSBMs with $p \neq q$ having $r$ equally-sized communities with indicator vectors $\boldsymbol{c_1}, \boldsymbol{c_2}, \ldots, \boldsymbol{c_k}$, expected adjacency matrix $\boldsymbol{A}$, and feature vector $\boldsymbol{x} = \boldsymbol{f} + \boldsymbol{\eta}$. Let $\boldsymbol{x}_{ASGC}$ be the feature vector returned by applying ASGC, with number of hops $K \geq r$, to $\boldsymbol{A}$ and $\boldsymbol{x}$. Then, $\boldsymbol{x}_{ASGC} = \boldsymbol{f} + \sum_i \theta_i' \boldsymbol{c_i}$, where each $\theta_i'$ is distributed according to $\mathcal{N}\left( 0, \tfrac{r}{n}\sigma^2 \right)$.*

*Proof.* Following the same argument from Section 5, the least squares in ASGC is equivalent to projecting the feature $\boldsymbol{x}$ onto the column span of the Krylov matrix $\boldsymbol{T} = \left( \boldsymbol{S}^1 \boldsymbol{x}; \boldsymbol{S}^2 \boldsymbol{x}; \ldots; \boldsymbol{S}^K \boldsymbol{x} \right)$. The column span of $\boldsymbol{T}$ is contained in the column span of $\boldsymbol{S}$, and since $\boldsymbol{S}$ is rank-$r$ (by the assumption that $p \neq q$), with probability 1 over the distribution of $\boldsymbol{\eta}$, as long as $K \geq r$, the column span of $\boldsymbol{T}$ equals that of $\boldsymbol{S}$; by Equation 5 for $\boldsymbol{S}$, this span is exactly that of the community indicator matrix $\boldsymbol{Q}$. Thus, ASGC is equivalent to multiplication of the feature $\boldsymbol{x}$ by the projection matrix $\boldsymbol{Q}\boldsymbol{Q}^\top$, for which we use Equation 6 as follows:

$$
\begin{aligned}
\boldsymbol{x}_{\text{ASGC}} &= \boldsymbol{Q}\boldsymbol{Q}^\top \boldsymbol{x} \\
&= \sum_i \left( \mu_i + \left( \sqrt{r/n} \right)\eta_i' \right) \boldsymbol{c_i} \\
&= \boldsymbol{f} + \sum_i \left( \sqrt{r/n} \right)\eta_i' \boldsymbol{c_i} \\
&= \boldsymbol{f} + \sum_i \theta_i' \boldsymbol{c_i},
\end{aligned}
$$

where $\theta_i' = \left( \sqrt{r/n} \right)\eta_i'$, which has the specified distribution. $\square$

## 9.2 Hyperparameter Study

We investigate the effect of two hyperparameter choices on validation accuracy: number of hops $K$ and regularization strength $R'$. We plot results for each hyperparameter in Figures 5 and 6, respectively. Trends vary significantly across datasets, indicating that tuning these hyperparameters is essential to optimal performance. This is perhaps to be expected for $R'$ in particular, since this hyperparameter controls, in a fairly direct way, the influence of the raw feature on the nodes' representations, whereas in an end-to-end method, the influence can be determined more indirectly based on information from the class labels.

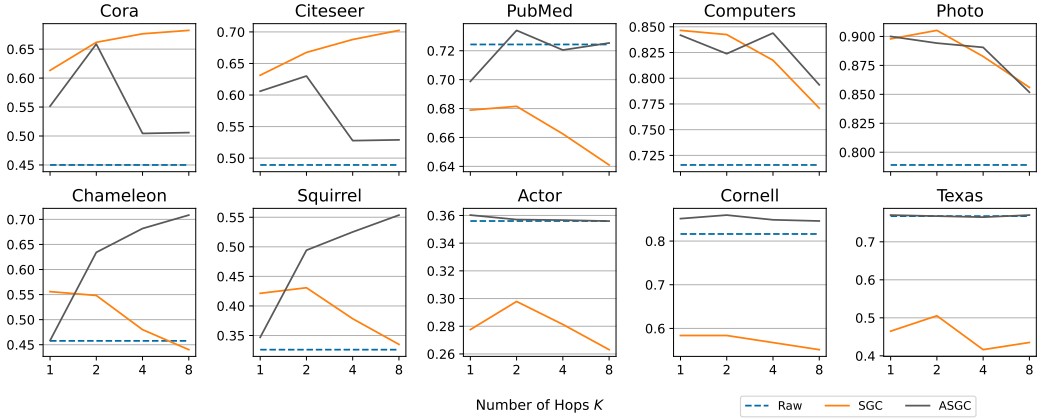

Figure 5: Validation accuracy for the non-deep methods as the number of hops $K$ is varied. Mean accuracy over 10 random splits. For ASGC, at each value of $K$, we report the regularization strength $R'$ with highest mean validation accuracy.

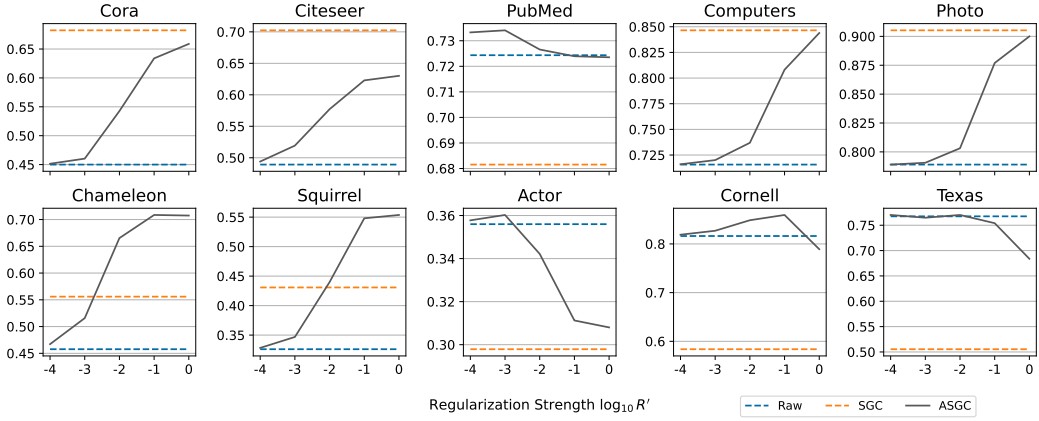

Figure 6: Validation accuracy for the non-deep methods as the regularization strength $R'$ is varied. Mean accuracy over 10 random splits. For ASGC, at each value of $R'$, we report the number of hops $K$ with highest mean validation accuracy.

## 9.3 Full Node Classification Results

Table 2 reports the full node classification performance results, which were deferred in Section 7. We provide results for our implementations of logistic regression on raw features, SGC, and ASGC; we also include results from Chien et al. (2021) for 9 deep methods, including their GPR-GNN. As the authors note, the results for Geom-GCN (Pei et al., 2020) on datasets not originally tested in that paper (in particular, the co-purchasing networks COMPUTERS and PHOTO) cannot be included due to a pre-processing subroutine that is not publicly available.

Table 2: Complete results for mean test classification accuracy, as well as 95% confidence intervals, on the benchmark of datasets from Table 1. Datasets are separated by homophilous vs heterophilous. Methods are separated by non-deep vs deep. Results for deep methods are taken from Chien et al. (2021).

| | CORA | CITESEER | PUBMED | COMPUTER | PHOTO | CHAMELEON | SQUIRREL | ACTOR | TEXAS | CORNELL |
|---|---|---|---|---|---|---|---|---|---|---|
| Raw | 55.09±1.81 | 60.30±1.55 | 77.79±0.95 | 76.07±0.57 | 82.97±0.58 | 49.56±0.88 | 34.16±0.74 | 36.28±0.77 | 86.49±2.88 | 86.49±2.88 |
| SGC | 78.16±1.32 | 70.18±1.00 | 73.90±2.22 | 87.14±0.45 | 92.03±0.51 | 57.70±1.62 | 44.98±1.28 | 30.07±0.76 | 55.68±5.71 | 54.32±6.41 |
| ASGC | 73.93±2.51 | 67.73±0.71 | 79.05±0.97 | 86.72±0.50 | 91.74±0.33 | 72.28±0.90 | 58.98±1.01 | 36.45±0.79 | 86.76±3.58 | 86.22±3.08 |
| MLP | 50.34±0.48 | 52.88±0.51 | 80.57±0.12 | 70.48±0.28 | 78.69±0.30 | 46.72±0.46 | 31.28±0.27 | 38.58±0.25 | 92.26±0.71 | 91.36±0.70 |
| GCN | 75.21±0.38 | 67.30±0.35 | 84.27±0.01 | 82.52±0.32 | 90.54±0.21 | 60.96±0.78 | 45.66±0.39 | 30.59±0.23 | 75.16±0.96 | 66.72±1.37 |
| GAT | 76.70±0.42 | 67.20±0.46 | 83.28±0.12 | 81.95±0.38 | 90.09±0.27 | 63.9±0.46 | 42.72±0.33 | 35.98±0.23 | 78.87±0.86 | 76.00±1.01 |
| SAGE | 70.89±0.54 | 61.52±0.44 | 81.30±0.10 | 83.11±0.23 | 90.51±0.25 | 62.15±0.42 | 41.26±0.26 | 36.37±0.21 | 79.03±1.20 | 71.41±1.24 |
| JKNet | 73.22±0.64 | 60.85±0.76 | 82.91±0.11 | 77.80±0.97 | 87.70±0.70 | 62.92±0.49 | 44.72±0.48 | 33.41±0.25 | 75.53±1.16 | 66.73±1.73 |
| GCN-Cheby | 71.39±0.51 | 65.67±0.38 | 83.83±0.12 | 82.41±0.28 | 90.09±0.28 | 59.96±0.51 | 40.67±0.31 | 38.02±0.23 | 86.08±0.96 | 85.33±1.04 |
| GeomGCN | 20.37±1.13 | 20.30±0.90 | 58.20±1.23 | NA | NA | 61.06±0.49 | 38.28±0.27 | 31.81±0.24 | 58.56±1.77 | 55.59±1.59 |
| APPNP | 79.41±0.38 | 68.59±0.30 | 85.02±0.09 | 81.99±0.26 | 91.11±0.26 | 51.91±0.56 | 34.77±0.34 | 38.86±0.24 | 91.18±0.70 | 91.80±0.63 |
| GPRGNN | 79.51±0.36 | 67.63±0.38 | 85.07±0.09 | 82.90±0.37 | 91.93±0.26 | 67.48±0.40 | 49.93±0.53 | 39.30±0.27 | 92.92±0.61 | 91.36±0.70 |