# OpenReview forum: "Simplified Graph Convolution with Heterophily"
_NeurIPS.cc/2022/Conference — NeurIPS 2022 Accept_

### Official Review · Reviewer_NDRL · 2022-07-09

**Rating:** 5
**Confidence:** 4
**Soundness:** 2 fair
**Presentation:** 3 good
**Contribution:** 2 fair

**Summary:**

This paper study the extension of SGC for heterophily settings. More specifically, it is known that SGC work for the homophilous setting where nodes that have similar labels are connected by an edge, however, SGC performs poorly on heterophily graph where nodes with different labels are connected by an edge.
The authors consider an multi-hop SGC as $\mathbf{H} = \sum_{i=0}^K \beta_i \mathbf{S}^i \mathbf{X},~\mathbf{S}=\mathbf{D}^{-1/2} \mathbf{A} \mathbf{D}^{-1/2}$.
To overcome the performance degradation issue for applying SGC on heterophily graphs, the authors propose an adaptive SGC method that learns the combination coefficient $\beta_i$ (as shown in Eq. 4).

**Questions:**

**Won't Eq. 4 with regularization lead to a degenerated solution?**

For example, I can write the Eq. 4 as
$$
\min_{\boldsymbol{\beta}} f(\boldsymbol{\beta}) = \sum_{i=1}^K \\| \beta_i (\mathbf{S}^i \mathbf{x} - \mathbf{x})\\|_2^2 + \mathbf{I}\\{i=0\\} R^2 \beta_0^2,
$$
where $S^0 = \mathbf{I}_N$ by using the authors' notation.
Taking gradient to each $\beta_i, i\in \{ 0,\ldots, K\}$ and set to zero, we have
$$
\frac{\partial f(\boldsymbol{\beta})}{\partial \beta_i} = 2\beta_i (\mathbf{S}^i \mathbf{x} - \mathbf{x}) (\mathbf{S}^i \mathbf{x} - \mathbf{x}) + \mathbf{I}\\{i=0\\} 2R^2 \beta_0 = 0
$$

Then, the closed form solution still has $\beta_0 \neq 0$ but $\beta_i = 0, i=1,\ldots,K$, which is the degenerated solution ...
in other words, the optimal $\boldsymbol{\beta}$ gives us linear NN.

**Strengths And Weaknesses:**

**Originality**

Both the method and the analysis in this paper sound novel and interesting to me.

**Quality + Significance**

I have doubts about the correctness of the adaptive SGC method in Eq. 4. Please refer to the Question section.

If I misunderstood the method in Eq. 4 and the author could correct me, then the contribution is significant. Otherwise, the authors might need to double-check their experiment results because according to my understanding, Eq. 4 should lead to a degenerated solution ...


**Clarity**

The paper is easy to follow and the intuition is quite clear.

---

> ### Author Response · Authors · 2022-08-01
> **Response to Reviewer NDRL**
>
> We appreciate that this review recognizes the novelty of the method and analysis. We address the sole criticism, which concerns soundness of the method.
>
> We agree that it is critical that our regularized loss function does not lead to a degenerate solution with $\beta_i =0$ for all $i>0$. However, we believe your rewriting of our loss function is incorrect.
>
> Ignoring the regularization part of the loss (your expression for this part matches ours), we have
> \begin{equation*}
> || \mathbf{T}\mathbf{\beta} - \mathbf{x} ||_2^2 = || \left( \sum \nolimits\_{i=0}^K \beta_i \mathbf{S}^i \mathbf{x} \right) - \mathbf{x} ||_2^2 \neq \sum \nolimits\_{i=0}^K || \beta_i \left( \mathbf{S}^i \mathbf{x} - \mathbf{x} \right) ||_2^2
> \end{equation*}
> where the left expression is as we wrote it in our paper, the right expression is what you wrote in your review, and the middle expression is a correct rewriting of our expression. Note that the middle and right expressions do not match because the sum cannot be taken outside the squared norm (and also because the multiplication by $\beta$ should not apply to the second $\mathbf{x}$).
>
> Based on our theoretical and experimental work, we are quite certain that our method does not present a degeneracy of this sort, and we would greatly appreciate it if you could update your review in light of this. If further clarification is needed during the discussion phase, please let us know and we would be happy to provide it. Thank you for the review.

---

> > ### Comment · Reviewer_NDRL · 2022-08-06
> > **Need more clarification**
> >
> > Let us suppose $\sum_i \beta_i = 1$ since it can be any number.
> >
> > $$
> > \min\_{\boldsymbol{\beta}} | \mathbf{T}\boldsymbol{\beta} - \mathbf{x} ||\_2 = \min_{\boldsymbol{\beta}} || \left( \sum_{i=0}^K \beta\_i \mathbf{S}^i \mathbf{x} \right) - \mathbf{x} ||\_2
> > $$
> >
> > $$= \min\_{\boldsymbol{\beta}} || \left( \sum_{i=0}^K \beta\_i \mathbf{S}^i \mathbf{x} \right) - \left(\sum\_{i=0}^K  \beta\_i\right) \mathbf{x} ||\_2  $$
> >
> > $$= \min\_{\boldsymbol{\beta}} || \left( \sum \nolimits\_{i=0}^K \beta\_i (\mathbf{S}^i \mathbf{x} - \mathbf{x})\right) ||\_2 $$
> >
> > $$\leq \min\_{\boldsymbol{\beta}} \sum \nolimits\_{i=0}^K || \beta\_i \left( \mathbf{S}^i \mathbf{x} - \mathbf{x} \right) ||\_2
> > $$
> >
> > It would be great if the authors could elaborate on why the optimal solution on the RHS is not the one for LHS?

---

> > > ### Author Response · Authors · 2022-08-08
> > > **Response to Clarifying Question**
> > >
> > > One cannot assume that $\sum_i \beta_i = 1$ since we are optimizing over all possible $\beta \in \mathbb{R}^{K+1}$. $\beta$ is not a fixed vector, and it does not have a fixed sum of entries. Changing the magnitude of $\beta$ can change the loss: for example, if we found a 'good' $\beta$ with low least-squares error, i.e., $\mathbf{x}_{\text{ASGC}} = \mathbf{T} \beta \approx \mathbf{x}$, then doubling $\beta$ would double $\mathbf{x}_\{\text{ASGC}}$, which could significantly increase error. Thus, the second line of the derivation is incorrect.
> > >
> > > Additionally, the fourth line is an inequality, so there is no reason to think these two optimization problems are equivalent. As a simple example, consider two functions, $f(x) = x^2$ and $g(x) = 2x^2 + x + 1$. $f(x) \leq g(x)$ for all $x$, but these functions have different minimizers.
> > >
> > > As a reminder, while we are examining your rewriting of the first term of our loss function, as is discussed in Lines 111-113 and 124-136, the regularization term $(R \beta_0)^2$ is what prevents the degenerate solution with $\beta_0 = 1$ and all other $\beta_i = 0$, which would yield $\mathbf{x}_{\text{ASGC}} = \mathbf{x}$. As $R$ is raised to higher values, this degenerate solution is clearly not minimizing the total loss, since the regularization term of the loss increases indefinitely.
> > >
> > > We appreciate your continued engagement, and we hope this discussion has resolved your concern. We reiterate that we are quite certain that our method does not present such a degeneracy, but please let us know if any further clarification is needed.

---

> > > > ### Comment · Reviewer_NDRL · 2022-08-08
> > > > **Need further clarification**
> > > >
> > > > Just let $\sum_i \beta_i = r, \forall r\neq 0$ and take gradient with respect to each $\beta_i$ and set to zero, won't we get the same conclusion as I first mentioned?
> > > > $\partial \\| \beta_i \left( \mathbf{S}^i \mathbf{x} - r \mathbf{x} \right) ||_2^2 / \partial \beta_i = \beta_i \left( \mathbf{S}^i \mathbf{x} - r \mathbf{x} \right)^\top \left( \mathbf{S}^i \mathbf{x} - r \mathbf{x} \right) = 0 \rightarrow \beta_i = 0$
> > > >
> > > > The example made by the authors is improper. In the inequality, I decompose the original function into multiple independent functions ($\beta_i$ and $\beta_j$ are two different variables for $i\neq j$). A proper example is $f(x) = (x_1 + x_2 + r)^2$, $g(x) = (x_1 + r/2)^2 + (x_2+r/2)$ ...

---

> > > > > ### Author Response · Authors · 2022-08-08
> > > > > **Reply**
> > > > >
> > > > > If $r \neq 1$ then the function $|| \sum \beta_i \mathbf{S}^i \mathbf x - r\mathbf{ x}||_2$ is not the same as the function that we are optimizing. Thus, is not clear why it is relevant to consider setting the derivative of this function to $0$. It is also not the case that separating the norm into a sum of individual components (which upper bounds the norm) and setting the derivative of each component to $0$ makes the derivative of the function itself equal to $0$. It just makes the derivative of the upper bounding function equal to $0$.
> > > > >
> > > > > We would like to stress that the optimization problem we are solving is nothing more than a regularized $\ell_2$ regression problem.
> > > > >
> > > > > For a simple example, let's assume we are in two dimensions, that $K = 1$, and that our regularization parameter $R = 1$. Assume that  $\mathbf x = [1,1]$, so $\mathbf S^0 \mathbf{x} = \mathbf{x} = [1,1]$. Further assume that $\mathbf{S}^1 \mathbf{x} = [1,0]$. Then our optimization problem is:
> > > > >
> > > > > $\min_{\beta_0, \beta_1} || [\beta_0, \beta_0] + [\beta_1, 0] - [1,1] ||_2^2 + \beta_0^2$.
> > > > >
> > > > > Expanding out the norm this is equal to $\min_{\beta_0, \beta_1} (\beta_0 + \beta_1 -1)^2 + (\beta_0-1)^2 + \beta_0^2$.
> > > > >
> > > > > We can check, e.g. via Wolfram Alpha (https://www.wolframalpha.com/input?i=minimum+of+%28z+%2B+w+-1%29%5E2+%2B+%28z-1%29%5E2+%2B+z%5E2) that this function is minimized when $\beta_0 = \beta_1 = 1/2$.

---

> > > > > > ### Comment · Reviewer_NDRL · 2022-08-08
> > > > > > **concern is addressed but need paper revision.**
> > > > > >
> > > > > > My concern is addressed and updated my score.
> > > > > >
> > > > > > Suggestions on revision:
> > > > > >
> > > > > > Take derivative with respect to $ \min_{\boldsymbol{\beta}} || \left( \sum_{i=0}^K \beta\_i \mathbf{S}^i \mathbf{x} \right) - \mathbf{x} ||\_2^2 + R\beta_0^2$ and set to zero, and show when $R$ is large enough (for example when $R>?$) $\beta_0=1, \beta_1,\ldots,\beta_K=0$ is not a solution.Because when $R$ is not large enough $\beta_1,\ldots, \beta_K\approx 0$ according to the closed-form solution.
> > > > > >
> > > > > > And also show why the decomposed solution I write above (for any $\sum_{i=0}^K \beta\_i = r$) is not identical to the solution for $ \min_{\boldsymbol{\beta}} || \left( \sum_{i=0}^K \beta\_i \mathbf{S}^i \mathbf{x} \right) - \mathbf{x} ||\_2^2 + R\beta_0^2$.
> > > > > >
> > > > > > Otherwise, I believe many readers will have the same doubt as I do when reading your problem formulation.

---

> > > > > > > ### Author Response · Authors · 2022-08-08
> > > > > > > **Reply**
> > > > > > >
> > > > > > > Thank you very much for taking the time to understand our work. We appreciate the suggestions on presentation.

---

### Official Review · Reviewer_86p6 · 2022-07-10

**Rating:** 4
**Confidence:** 4
**Soundness:** 3 good
**Presentation:** 3 good
**Contribution:** 3 good

**Summary:**

The paper introduces Adaptive SGC as an improvement of SGC to better handle heterophilous graphs. The major idea is to learn a polynomial of normalized adjacency matrix by solving a least square problem. The proposed idea is simple and scalable. Theoretical analyses under very ideal assumptions are provided. The empirical performance is competitive with recent baselines.

**Questions:**

Please respond to the comments mentioned in the weakness.

In addition to that, could you please compare the advantages and disadvantages of the proposed filter learning and existing filter learning such as ChebNet and GPR-GNN in terms of accuracy regardless of efficiency？

**Strengths And Weaknesses:**

Strengths:

1. The paper introduces a simple and scalable approach for graph learning in both homophilous and heterophilous settings.

2. The paper is clearly written and easy to follow.

3. Motivation examples and theoretical analyses are provided to justify the proposed idea.

4. Empirical evidence supports the effectiveness of the proposed algorithm.

Weakness:

1. Despite the motivation example and theoretical analyses, the intuition and motivation of the proposed filter learning via lease square approximation are still unclear. There is a lack of discussion about the insight of projecting the feature onto the column space of the Krylov matrix.

2. The motivation example and theoretical analyses are provided under ideal assumptions. For instance, the adjacent matrix is the expected adjacent matrix that precisely reflects the membership information of each node. It is unclear how the sampled setting impacts the current theoretical results.

3. The theoretical analysis removes the raw feature in the least square approximation. But the impact of raw feature and coefficient regularization are never studied in the experiments.

4. It is unclear why ASGC would achieve similar performance as using the raw feature. A shallow discussion is provided in Section 3 but it is not sufficient. A further discussion about when and why the proposed method fails will also be helpful to understand its limitation.

5. The performance improvements over recent baselines are not stable and are sometimes worse.

---

> ### Author Response · Authors · 2022-08-01
> **Response to Reviewer 86p6**
>
> Thank you for the review. We address some points of concern below.
>
> >the intuition and motivation of the proposed filter learning via least square approximation are still unclear
>
> >The theoretical analysis removes the raw feature in the least square approximation. But the impact of raw feature and coefficient regularization are never studied in the experiments.
>
> >It is unclear why ASGC would achieve similar performance as using the raw feature.
>
> As you note, we provide some intuition for the method in Section 3; we also discuss a more involved interpretation in the appendix (Section A.1). Briefly, concerning how in some situations ASGC would theoretically achieve similar performance as the raw feature: ASGC finds and returns a linear combination of the raw feature $\mathbf{x}$ and some other vectors, attempting to match the target vector $\mathbf{x}$ itself via least-squares. A trivial solution with 0 error is to put a coefficient of 1 on $\mathbf{x}$ and 0 on the other vectors. We avoid this via penalizing/regularizing the coefficient on the vector $\mathbf{x}$ in the linear combination. Hence, in the low penalty limit, we simply return the trivial solution (i.e., the raw vector), and the ASGC-filter deviates more from this trivial solution the higher the penalty. (Another possibility is if there are too many vectors to linearly combine; in this case also, the target $\mathbf{x}$ is too easily approximated, though this does not happen in practice.) The theory results are in the high penalty limit, while practically we set the penalty via cross-validation. The impact of varying the penalty strength is as expected from regularization in general: there is some “sweet-spot” of regularization strength where the maximum performance is achieved. We would include and discuss these results in a full version if the reviewer believes they would be of interest.
>
> >The performance improvements over recent baselines are not stable and are sometimes worse.
>
> >compare the advantages and disadvantages of the proposed filter learning and existing filter learning such as ChebNet and GPR-GNN in terms of accuracy regardless of efficiency？
>
> Indeed, ASGC does not achieve the highest performance of the surveyed methods across all datasets, though we find that it does achieve overall the best performance when aggregated across the 10 datasets (Figure 4), and it outperforms even GPR-GNN in terms of minimum accuracy. Even excluding our method, none of the other 11 methods achieve the best performance on all datasets. While there are similarities between our method and existing filter learning methods (we specifically compare and contrast with GPR-GNN in Lines 269-273), there are significant differences as we discuss in the overall response. Primarily, the existing filter-based methods, along with most other prior methods, use the common framework of end-to-end optimization of a neural network with backpropagation of a cross-entropy loss using the node labels. Our method lies outside this framework.
>
>
> Thank you again for your review. Please let us know if you have any further questions, and we will be happy to answer.

---

> > ### Comment · Reviewer_86p6 · 2022-08-09
> > **Reply**
> >
> > Dear authors,
> >
> > Thank you for the response. However, the response does not fully address my major concerns:
> >
> > 1. You basically reinterpret what the lease square problem is solving but do not explain the intuition and motivation of the least square approximation.
> >
> > 2. Moreover, the theoretical analysis of the proposed idea is inconsistent with the algorithm being implemented. For instance, the raw feature is removed in the theory. It is expected to see a comprehensive study on the impact of raw feature and coefficient regularization in the experiments.
> >
> > 3. I do not see clear advantages of the proposed filter learning over existing filter learning (such as the polynomial filter).
> >
> > Reviewer 86p6

---

> > > ### Author Response · Authors · 2022-08-09
> > > **Responses to Reply**
> > >
> > > Thank you for your continued engagement. We address each of these three points:
> > >
> > > 1. In short, ASGC works by replacing the fixed feature propagation step of SGC with an adaptive one, which may or may not be smoothing based on the feature and graph; this corresponds to learning (rather than asserting) a polynomial of the normalized adjacency matrix $\mathbf{S}$ which multiplies the feature $\mathbf{x}$ to filter it (Lines 105-108). We agree that this is merely a description of the capacity of ASGC to adapt to homophily/heterophily rather than a description of the precise mechanism by which the least squares learns polynomial coefficients that adapt (though we do discuss intuitively how the least squares might pick up positive/negative correlations with various propagated versions $\mathbf{S}^i \mathbf{x}$ of the feature $\mathbf{x}$ in Lines 130-134). However, the latter kind of description is generally not provided by most prior works either, e.g., the deep filter learning methods do not describe the exact mechanism by which cross-entropy SGD updates learn suitable coefficients; rather, they describe the new layer/component they add to the network and how it has this capacity. Some interesting recent work ("A Unifying Generative Model for Graph Learning Algorithms") has shown that models close to SGC can be derived as MLE when asserting a certain graph generative model which assumes homophily; it would be interesting to show some similar result for ASGC with a more general generative model, but this is a significant contribution beyond our work.
> > >
> > > 2. There are two limits of regularization: $R \to \infty$, in which case the raw feature is not present in the filter at all ("pure" ASGC), and $R \to 0$, in which case the filter does nothing and simply returns the raw feature ("Raw"). Tuning $R$ allows ASGC to interpolate between these extremes in a way that is specific to each dataset. We now include links to validation accuracy versus [$\log_{10}(R')$](https://i.ibb.co/xqk3ktp/image.png) (which increases with $R$, see Lines 296-297), as well as number of hops [$K$](https://i.ibb.co/f2zxRzz/image.png) (note that these are over an overlapping but slightly different set of networks). We would be happy to update, include, and discuss these results if the reviewer believes it is constructive. The theoretical results assume the former limit ("pure" ASGC) - indeed this is not the setting we use for experiments, but we note that provably showing denoising in both homophilous/heterophilous settings, even with the strong assumptions, is to our knowledge a new result.
> > >
> > > 3. While there may be some ease of implementation, efficiency, and other practical advantages depending on the use case, which we have discussed in the paper and other responses, we principally emphasize the scientific value of this work. The mechanism by which the filter is learned is totally different to that of the other methods, but it is still competitive; we believe this has significance for the science of machine learning on graphs beyond improved measurements of accuracy/runtime.

---

### Official Review · Reviewer_MRH6 · 2022-07-11

**Rating:** 3
**Confidence:** 4
**Soundness:** 2 fair
**Presentation:** 3 good
**Contribution:** 2 fair

**Summary:**

This paper presents an adaptive simple graph convolution method (ASGC) for learning over heterophilous graphs. ASGC is proposed as an extension of the previously proposed simple graph convolution (SGC) to better handle graph heterophily. The authors provide a theoretical analysis of synthetic datasets to demonstrate the ineffectiveness of SGC on heterophilous graphs. Empirical comparisons with several state-of-the-art methods show some promising results of ASGC as compared with deep learning models.

**Questions:**

1. What is the key difference between the proposed ASGC method and Generalized PageRank GNN (GPR-GNN), except for the loss function? Why ASGC works better than GPR-GNN on some datasets, but worse on some other datasets?

2. The authors claim that they are the first to handle graph heterophily using feature pre-processing. What is the significance of pre-processing features to handle heterophily without considering any particular learning tasks? Please give more explanations.

3 Please give more explanations for the theoretical analysis provided. For example, what is the connection between the learned coefficients \beta and homophily ratio of graphs?



**Limitations:**

I think the authors have adequately addressed the limitations of their proposed method.

**Strengths And Weaknesses:**

Strengths:
1. This paper is well written and easy to follow.
2. The theoretical analysis based on stochastic block models show the limitations of SGC and the improvements of ASGC over SGC.

Weakness:
1. Technical contributions of this paper are somehow limited. In essence, I think the proposed method uses a similar "adaptive" idea as  Generalized PageRank GNN (GPR-GNN) -- which achieves adaptive smoothing over edges via learnable coefficients, thus better generalising to heterophilous graphs. It is unclear to me what advantages ASGC offers over GPR-GNN, apart from being a non deep learning model. Empirical results also show that ASGC is less effective than GPR-GNN on 3 out of 5 heterophilous graphs.

2. The authors claim that "our work is the first to show that heterophily can be handled using just feature pre-processing". I think this is overclaimed because the authors are mainly concerned about node classification on graphs. In a general sense, how to effectively handle heterophily is indeed dependent on the learning task to be addressed.

3. The experimental evaluation is weak. First, heterophilous graphs used in the experiments are quite small. Adding more and larger datasets would make the evaluation stronger. Second, empirically, ASGC is claimed to be competitive with recent deep learning models on heterophilous graphs. However, its performance is found to be inferior to GPR-GNN on more than half of heterophilous graphs. Third, the authors have not provided comparisons with several recent works for handling graph heterophily.

---

> ### Author Response · Authors · 2022-08-01
> **Response to Reviewer MRH6**
>
> Thank you for the review. We appreciate the concerns raised and address them below.
>
> >What is the key difference between the proposed ASGC method and Generalized PageRank GNN (GPR-GNN), except for the loss function?
>
> >What is the significance of pre-processing features to handle heterophily without considering any particular learning tasks?
>
> We discuss some differences in Lines 268-273: mainly, 1) GPR-GNN’s end-to-end model involves an initial transformation by a neural network, which is not present in our method, and 2) the training algorithm is the traditional cross-entropy backprop using node labels vs. our novel feature pre-processing method based on least-squares. We do not believe (2) is a minor difference. As we discuss in the overall response, this difference could have some practical implications, but more importantly, it is of scientific interest: to our knowledge, almost all, if not all, works claiming to handle node classification with heterophily involve deep methods and end-to-end backprop using labels. That our work is generally competitive with these methods advances our understanding of exactly what learning components are responsible/necessary for good performance on this task.
>
> >Why ASGC works better than GPR-GNN on some datasets, but worse on some other datasets?
>
> In general, it is difficult to explain such differences in performance, but there might be a relatively simple explanation in this case.
> In Lines 313-317, we note that, interestingly, the datasets on which GPR-GNN significantly outperforms ASGC are exactly those on which an MLP significantly outperforms logistic regression; the latter two methods ignore the graph, suggesting that some nonlinear processing of the node features may be key to performance on these networks, separate from how the graph is exploited.
> We do note that ASGC overall outperforms GPR-GNN and the other 10 methods when aggregating results over the 10 datasets (Figure 4).
>
> >heterophilous graphs used in the experiments are quite small. Adding more and larger datasets would make the evaluation stronger… the authors have not provided comparisons with several recent works for handling graph heterophily.
>
> We include results for 10 datasets, which are an (often strict) superset of datasets used in other recent works (including the GPR-GNN paper), and for 12 methods. This is an area of active research with frequent publications; as we state in Lines 338-341, we do not assert absolute state-of-the-art results, and we believe the significance of this work is more based on the uniqueness of the method and the scientific implications of its effectiveness.
>
> >what is the connection between the learned coefficients \beta and homophily ratio of graphs?
>
> The coefficients of $\mathbf{\beta}$ may be a bit arbitrary since they correspond to the specific nonlinear basis $\mathbf{S}\mathbf{x},\mathbf{S}^2\mathbf{x},\dots,\mathbf{S}^K\mathbf{x}$ (the Krylov subspace). This is why we instead analyze in terms of the basis of eigenvectors of $\mathbf{S}$. Specifically, the two eigenvectors are an all-ones vector (corresponding to the average of the two communities’ mean feature values), and a vector indicating the two communities in its signs (corresponding to the difference between the two communities’ mean feature values). We show that in ASGC, the coefficients on both are fixed (we simplify this argument by using the fact that the least-squares in ASGC is just a projection onto the basis of $\mathbf{S}$), whereas in SGC, the coefficient for the latter shifts with the homophily value $\lambda_2$, which can have undesirable effects.
>
> >The authors claim that "our work is the first to show that heterophily can be handled using just feature pre-processing". I think this is overclaimed because the authors are mainly concerned about node classification on graphs.
>
> Most recent works about heterophily mainly seem to concern node classification, but this is a fair point. We will revise this sentence to “…first to show that node classification with heterophily can be handled…”
>
>
> Thank you again for your review, and please let us know if you have any further questions.

---

### Official Review · Reviewer_X7Nw · 2022-07-12

**Rating:** 6
**Confidence:** 3
**Soundness:** 3 good
**Presentation:** 2 fair
**Contribution:** 3 good

**Summary:**

This paper proposes Adaptive Simple Graph Convolution (ASGC), which can adapt to both homophilous and heterophilous graph structures. ASGC represents an extension of SGC to graph with heterophily, which is a novel and timely contribution.

**Questions:**

Suggestions:
- Typos on Figure 3. Should be ASGC, not AGC.

Questions:
- Could ASGC extend to graph classification tasks?

**Limitations:**

Yes.

**Strengths And Weaknesses:**

Strengths:
- The contribution is novel. Most existing methods for simplifying Graph Neural Networks make the homophily assumption. This paper extends the use case of this line of work to heterophilous graphs, which have wide applications in cases like networks based on adjacencies of words in text
- The paper provides nice theoretical guarantees on the capability of proposed ASGC at denoising FSBM networks.

Weakness:
- More explorations regarding the potential of ASGC are needed. Based on the experimental results in Figure 3, ASGC actually performs worse than the standard SGC in a few datasets. I hope the authors can provide more justifications and analyses on why this is the case.
- I hope the paper can somehow unify the idea of ASGC and SGC, such that a model can automatically decide and apply the best feature propagation scheme based on the task of interest. I.e., the model tends to be more ASGC alike when the input graph is more heterophilous, and the model is more SGC alike when the input graph is more homophilous. This improvement to the model should make it much easier to use, as a user no longer has to decide between ASGC and SGC.

---

> ### Author Response · Authors · 2022-08-01
> **Response to Reviewer X7Nw**
>
> Thank you for the review. We appreciate the recognition of the method’s novelty and theoretical work, and we address the points of concern below.
>
> >Based on the experimental results in Figure 3, ASGC actually performs worse than the standard SGC in a few datasets. I hope the authors can provide more justifications and analyses on why this is the case.
>
> Indeed, on four of the five homophilous datasets, SGC outperforms ASGC. We believe this is not entirely unexpected. While ASGC can adapt to both homophily and heterophily, SGC assumes homophily; this assumption can be problematic, but when the dataset matches this assumption, SGC has a useful inductive bias and can perform very well, similar to how one might expect linear regression to outperform quadratic regression when the underlying relationship is linear. The experiments on synthetic datasets in Section 4 also illustrate this.
> As an aside, we note that SGC also outperforms the majority of baselines on these datasets, often by a much larger margin than it outperforms ASGC (Table 2), and that ASGC overall outperforms SGC and the other 10 methods when aggregating results over the 10 datasets (Figure 4).
>
> >I hope the paper can somehow unify the idea of ASGC and SGC, such that a model can automatically decide and apply the best feature propagation scheme based on the task of interest.
>
> This is an interesting idea, e.g., use cross-validation to choose between SGC and ASGC. We have more generally explored the idea of using a convex combination of raw, SGC-filtered, and ASGC-filtered features, where the optimal combination weights are found by cross-validation. This works well empirically, and we believe it could indeed simplify use of these methods in practice. We have only avoided including this combination method here to avoid confusion about our paper’s message (i.e., the message that it is possible to adapt to both homophily/heterophily with non-deep feature preprocessing).
>
> >Could ASGC extend to graph classification tasks?
>
> This is an interesting question. Yes, we believe so. For example, ASGC could be used to proprocess features, then the node features could be pooled to generate graph features for graph classification. This highlights one strength of ASGC - node classes would not be required for this procedure (though they could be integrated as extra features if available).
>
>
> If any more questions arise during the discussion period, we will be happy to address them. Thank you again.

---

### Author Response · Authors · 2022-08-01
**Overall Response to Reviews**

We thank the reviewers for their comments. We would briefly like to highlight what we believe is the main contribution of our work, which may have been underemphasized. Most recent methods for node classification (with a notable exception being SGC), and virtually all, if not all methods for node classification in a heterophilous setting, are based on a certain popular framework: end-to-end optimization of a neural network with backpropagation of a cross-entropy loss using the node labels. This naturally raises a scientific question: are these components necessary for good performance in this setting? Our work proposes a novel alternate method outside this framework: a feature pre-processing method based on least-squares. Therefore, we believe the competitive performance of our method advances our understanding of this task regarding exactly what learning components are responsible/necessary for good performance on this task. Practically, methods like ours may have benefits in some pipelines where labels are not available upfront or end-to-end learning is otherwise problematic, and the simplicity of implementing the method may also be appealing; however, we principally emphasize the scientific value of this work. The contribution of the SGC paper was similar, and it has had a significant impact.

---

### Meta-Review · Area_Chair_WkTL · 2022-08-29

**Recommendation:** Accept
**Confidence:** Certain

**Metareview:**

This paper proposes a new feature smoothing graph learning algorithm handling heterophilous graphs. The proposed idea is an adaptive node feature smoothing technique which essentially uses a regularized least square to find a weighted sum of the Krylov matrix columns (smoothed features at different orders) as an approximation of the original features. Careful theoretical and empirical analyses are carried out on a stochastic block model with 2 communities. These analyses, although in a rather restrictive setting, provide valuable insights.

Most reviewers appreciated the novelty of the idea and appreciated the theoretical and empirical insights on the 2-cluster SBM setting. The benefit over previous method (SGC) on heterophilous graphs are well established. These feature-smoothing methods are generally interesting and important; they can better reveal the mechanism of graph learning and potentially improve efficiency and representation learning. Using non-deep Most technical concerns (formulation, evaluation, baselines, datasets) are addressed and clarified during rebuttal and discussion period. The theoretical results, although restricted to a relatively limited setting, provide a good intuition of how the idea works and help build good heuristics in real world setting. This is valuable and will benefit NeurIPS audience.


**Award:**

No

---

### Decision · Program_Chairs · 2022-09-14

Accept